# Mapping the spatial transcriptomic signature of the hippocampus during memory consolidation

Yann Vanrobaeys[1,2,3], Utsav Mukherjee [1,2,4], Lucy Langmack[1,2,5], Stacy E. Beyer[1,2], Ethan Bahl[3,6], Li-Chun Lin[1,2], Jacob J. Michaelson [2,6], Ted Abel [1,2] ✉ & Snehajyoti Chatterjee [1,2] ✉

Memory consolidation involves discrete patterns of transcriptional events in the hippocampus. Despite the emergence of single-cell transcriptomic profiling techniques, mapping the transcriptomic signature across subregions of the hippocampus has remained challenging. Here, we utilized unbiased spatial sequencing to delineate transcriptome-wide gene expression changes across subregions of the dorsal hippocampus of male mice following learning. We find that each subregion of the hippocampus exhibits distinct yet overlapping transcriptomic signatures. The CA1 region exhibited increased expression of genes related to transcriptional regulation, while the DG showed upregulation of genes associated with protein folding. Importantly, our approach enabled us to define the transcriptomic signature of learning within two less-defined hippocampal subregions, CA1 stratum radiatum, and oriens. We demonstrated that CA1 subregion-specific expression of a transcription factor subfamily has a critical functional role in the consolidation of long-term memory. This work demonstrates the power of spatial molecular approaches to reveal simultaneous transcriptional events across the hippocampus during memory consolidation.

Activity-dependent gene expression occurs in wave-like patterns following experience. The early wave of transcriptional events involves increased expression of immediate early genes (IEGs) and newly synthesized proteins to regulate downstream gene expression[1–3]. IEGs encoding transcription factors, such as *Fos, Egr1*, and the *Nr4a* subfamily, regulate a larger, more diverse set of effector genes that mediate the structural and functional changes underlying synaptic plasticity. Gene expression at these critical time points is essential to drive responses to experience, including memory consolidation. Newly formed memory is thought to be stored within functionally connected neuronal populations in the hippocampus through dynamic gene expression patterns, known as engram ensembles[4–7].

Dynamic transcriptional patterns within the circuitry supporting engram ensembles enable the consolidation of memory from the hippocampus to multiple brain regions[4,8–11]. Neuronal populations contributing to engram ensembles are activated by learning and endure cellular changes[10,12], which can later be reactivated for memory retrieval[13] or inhibited inducing memory impairments[14]. Therefore, understanding the transcriptional dynamics throughout the hippocampus following an experience would provide important insights into the molecular mechanisms underlying memory consolidation.

The circuitry encompassing different subregions of the dorsal hippocampus has distinct roles in memory consolidation[15–17]. Layer II of the entorhinal cortex (EC) projects to granule cells of the dentate gyrus

[1]Department of Neuroscience and Pharmacology, Carver College of Medicine, University of Iowa, Iowa City, IA, USA. [2]Iowa Neuroscience Institute, University of Iowa, Iowa City, IA, USA. [3]Interdisciplinary Graduate Program in Genetics, University of Iowa, Iowa City, IA 52242, USA. [4]Interdisciplinary Graduate Program in Neuroscience, University of Iowa, Iowa City, IA 52242, USA. [5]Biochemistry and Molecular Biology Graduate Program, University of Iowa, Iowa City, IA, USA. [6]Department of Psychiatry, University of Iowa, Iowa City, IA, USA. ✉e-mail: ted-abel@uiowa.edu; snehajyoti-chatterjee@uiowa.edu

(DG) and pyramidal neurons of the CA3 region through the perforant pathway (PP), and layer III of EC projects to the pyramidal neurons of CA1 through the temporoammonic and alvear pathways[18–20]. The direct EC input to CA1 is essential for spatial memory consolidation and novelty detection[21–24]. DG granule cells project onto CA3 pyramidal neurons through mossy fibers, and CA3 pyramidal neurons send projections to CA2 and CA1 pyramidal neurons through the Schaffer collateral (SC) pathway[25–27]. The axons from CA1 pyramidal neurons project onto subiculum and EC neurons, forming the major output pathway of hippocampal circuits[28]. The DG is the site of adult neurogenesis in the hippocampus[29]. Adult newborn granule cells mediate pattern separation in the DG[30], while mature granule cells in DG and CA3 pyramidal neurons are essential for pattern completion, involving associative memory recall from a partial cue[31,32]. Thus, hippocampal memory relies on the association between items and contexts[33], with neurons in the CA1 processing information about objects and locations[34] and DG neurons driving pattern separation to reduce overlap between neural representations of similar learning experiences[35–37]. Nevertheless, the spatial transcriptomic changes in response to learning underlying the circuitry across subregions of the dorsal hippocampus remain largely unknown.

Hippocampal engram ensembles have been studied using the expression of individual IEGs[11] within the whole hippocampus[38,39], CA1[40,41], DG[42,43], and hippocampal neuronal nuclei[41,44,45], but not across all subregions simultaneously. A relatively new approach is targeted recombination of active neuronal populations (TRAP) to study unbiased cell-type specific gene expression in the hippocampus following a learning experience[4,45]. *Fos* is one IEG that is thought to link hippocampal engrams and place codes underlying spatial maps[7,46]. Single-nuclei RNA sequencing identified downstream targets of *Fos* in CA1 pyramidal cells following neuronal stimulation[47] and defined the role of cell type-specific expression of *Fos* in CA1 for spatial memory[7,46,47]. Single-nuclei transcriptomic studies from Fos+ (activated) and Fos- (non-activated) hippocampal neurons following exposure to a novel environment revealed transcriptomic differences between DG and CA1 neurons[48]. Other studies have applied a similar approach in the hippocampus to capture engram cells following learning[45] or activated neurons following neuronal stimulation[2,43]. These advancements allow transcriptional profiles to be sorted into cell types based on canonical marker genes[49,50]. However, it is still unclear how gene expression differs after learning across each subregion without using engram-specific or IEG markers. Understanding unbiased expression will help define engram ensembles and identify unique roles for each of these subregions in memory consolidation.

Utilizing the spatial coordinates within intact brain tissue enables unbiased yet precise identification of transcriptomic changes at high spatial resolution[51,52]. Spatial transcriptomics combines both histology and spatial profiling of RNA expression to provide high-resolution transcriptomic characterization of distinct transcriptional profiles within individual brain subregions[53]. We have recently used the spatial transcriptomic approach to demonstrate neuronal activation patterns within brain regions using a deep-learning computational tool[54]. In this work, we have applied this state-of-the-art approach to examine activity-driven spatial transcriptomic diversity within the hippocampal network. We define genome-wide transcriptomic changes in the CA1 pyramidal layer, CA1 stratum radiatum, CA1 stratum oriens, CA2 + 3 pyramidal layer, and dentate gyrus (DG) granular and molecular layers of the dorsal hippocampus within the first hour following spatial exploration. Moreover, we demonstrated the functional relevance of our findings by selectively manipulating the function of the Nr4a subfamily of transcription factors within CA1 pyramidal neurons. Mapping the precise learning-induced expression pattern of genes across the hippocampus enhances our understanding of their subregion-specific role in memory consolidation.

## Results

### Pseudobulk analysis of hippocampal spatial transcriptomics following learning correlates with bulk RNA sequencing

The growing knowledge of transcriptomic heterogeneity in the hippocampus raises the critical question of which genes are selectively regulated within each subregion during a critical early time point of memory consolidation. We performed spatial transcriptomic analyses using the *10x Genomics Visium* platform in coronal brain slices obtained from adult C57BL/6 J male mice 1 hr after training in a hippocampus-dependent learning task compared to homecage controls (Spatial object recognition task, SOR, *n* = 4/group, Fig. 1a). We and others have previously demonstrated that the learning-induced early wave of gene expression peaks at this timepoint after learning[39,55,56]. We analyzed additional transcriptomic profiles by integrating our previous spatial transcriptomics dataset following SOR training[54] (GEO GSE201610, *n* = 3/group). Increasing the number of biological replicates can improve statistical power and improve the robustness of the results as shown previously[57,58]. The profiles used for all spatial transcriptomics data analyses (pseudobulk analysis, total *n* = 7/group, Supplementary Fig. 1) include the hippocampal subregions CA1 pyramidal layer, CA1 stratum radiatum, CA1 stratum oriens, CA2 and CA3 pyramidal layers and DG granular layers (Fig. 1b). Here, pseudobulk analysis refers to grouping of each dot from a given brain slice to form a single pseudo-sample. Differential gene expression analysis of this pseudobulk data revealed 119 differentially expressed genes (DEGs), with 101 upregulated and 18 downregulated genes following learning (Fig. 1c, d, Supplementary Data 1). The gene expression changes across the two datasets correlated significantly (Supplementary Fig. 2). Enrichment network analysis was used to identify the pathways most represented among the DEGs. The upregulated pathways include nuclear receptor activity, nucleotide transmembrane transporter activity, protein kinase inhibitor activity, dioxygenase activity and histone demethylase activity (Fig. 1e). The nuclear receptor activity includes the genes *Nr4a1*, *Nr4a2* and *Nr4a3*, that comprise a subfamily of transcription factors known to be involved in learning and memory[59,60]. Histone demethylation activity has been linked to memory consolidation[61], while mutations in JMJD1C are associated with intellectual disability[62]. Protein kinase inhibitors are often found to be upregulated following learning, acting as a negative regulators of transcription activation pathways, such as MAPK pathway[63], and memory suppressor genes[64]. Other IEGs upregulated following learning include *Egr1*, *Arc*, *Homer1*, *Per1*, *Dusp5*, and *Junb* and are all associated with learning and memory[2,38,65].

Over the past decade, bulk RNA sequencing (RNA-seq) has been extensively used to study transcriptional profiles from brain tissue[40,41,59]. Therefore, to complement our spatial transcriptomic approach with conventional transcriptomic tools, we performed RNA-seq using whole dorsal hippocampus tissue (bulk RNA-seq) from mice trained in SOR (1 h) or homecage. Bulk RNA-seq analysis revealed 224 DEGs (Fold change: 1.4, FDR < 0.05) following SOR training compared to control mice, with 147 upregulated and 77 downregulated genes after learning (Fig. 2a, Supplementary Data 2). We next asked whether our pseudobulk spatial transcriptomics data overlapped with learning-induced gene expression changes of bulk RNA-seq. Among the 101 upregulated genes from pseudobulk spatial transcriptomics, 29 genes were identified with bulk RNA-seq. Only one gene among 18 downregulated genes appeared in bulk RNA-seq. Genes differentially expressed in pseudobulk RNA-seq significantly correlate with bulk RNA-seq, and the directionality of the change in expression was maintained (Fig. 2b). *Nr4a1*, *Dusp5*, *Arc*, *Sgk1* appeared among the top genes both in the pseudobulk and the bulk RNA-seq (Fig. 2b, c). Among these genes, *Nr4a1* and *Sgk1* are upregulated within the 30 min-1 h temporal window and their expression is comparable to homecage post 1 h timepoints (Fig. 2d). Oligodendrocyte differentiation-related gene *Opalin* was the only common downregulated gene (Fig. 2b).

Pseudobulk analysis also revealed differentially expressed genes that were not identified by bulk RNA-seq approach. Some of these upregulated transcripts identified using pseudobulk spatial transcriptomics include genes related to chromatin binding (*Ncoa2, Polg, Smc3,* Bcl6, *Jdp2,* Sp3), protein kinase inhibitors activity (*Spred1, Trib2*) and chaperone binding (*Dnajc3, Sacs, Grpel2*). Some of the down-regulated genes included myelin oligodendrocyte glycoprotein (*Mog*), myelin-associated glycoprotein (*Mag*) and long noncoding RNA, *Mir9-3hg*. These results demonstrate that spatial transcriptomics using the Visium platform detects DEGs that overlap with other transcriptomic approaches yet reveals genes that may be undetectable in other techniques. On the contrary, bulk RNA-seq identified genes that were not identified in the pseudobulk analysis. Some of these include genes related to transcription regulation (*Fos, Egr2, Hif3a, Fosl2, Nfkbia*), protein processing in the Endoplasmic Reticulum (H*spa1b, Hspa1a, Herpud1, Pdia6, Pdia4, Hsph1*) and protein kinase regulator activity (*Hspb1, Cdkn1a, Trib1, Cables1*) (Supplementary Data 1 and 2). Thus, both the techniques can be applied to understand transcriptomic changes across hippocampal subregions.

## Hippocampal subregions exhibit distinct transcriptomic signatures following learning

The dorsal hippocampus is composed of multiple anatomically and functionally distinct subregions. Here we distinguished the major principal neuronal layers and memory-relevant hippocampal regions: CA1 pyramidal layer, CA1 stratum radiatum, CA1 stratum oriens, CA2 and CA3 pyramidal layers combined, and DG granular layer based on spatial topography by H&E staining (Fig. 3a). Computational analysis of the transcriptomic profiles from these hippocampal subregions reveals distinct clusters in a UMAP plot (Fig. 3b). Analyzing the hippocampal subregion-specific transcriptomic signature after learning revealed 58 differentially expressed genes in the CA1 pyramidal layer, 16 genes in the CA2 and CA3 pyramidal layers, and 104 genes in the DG molecular and granular layer (Supplementary Fig. 3, Supplementary Data 3). Within each subregion, spatial learning resulted in 46 upregulated and 12 downregulated genes in the CA1 pyramidal layer, 13 upregulated and 3 downregulated genes in CA2 and CA3 pyramidal layers, and 68 upregulated and 36 downregulated genes in DG (Fig. 3c). In addition to the CA1 pyramidal

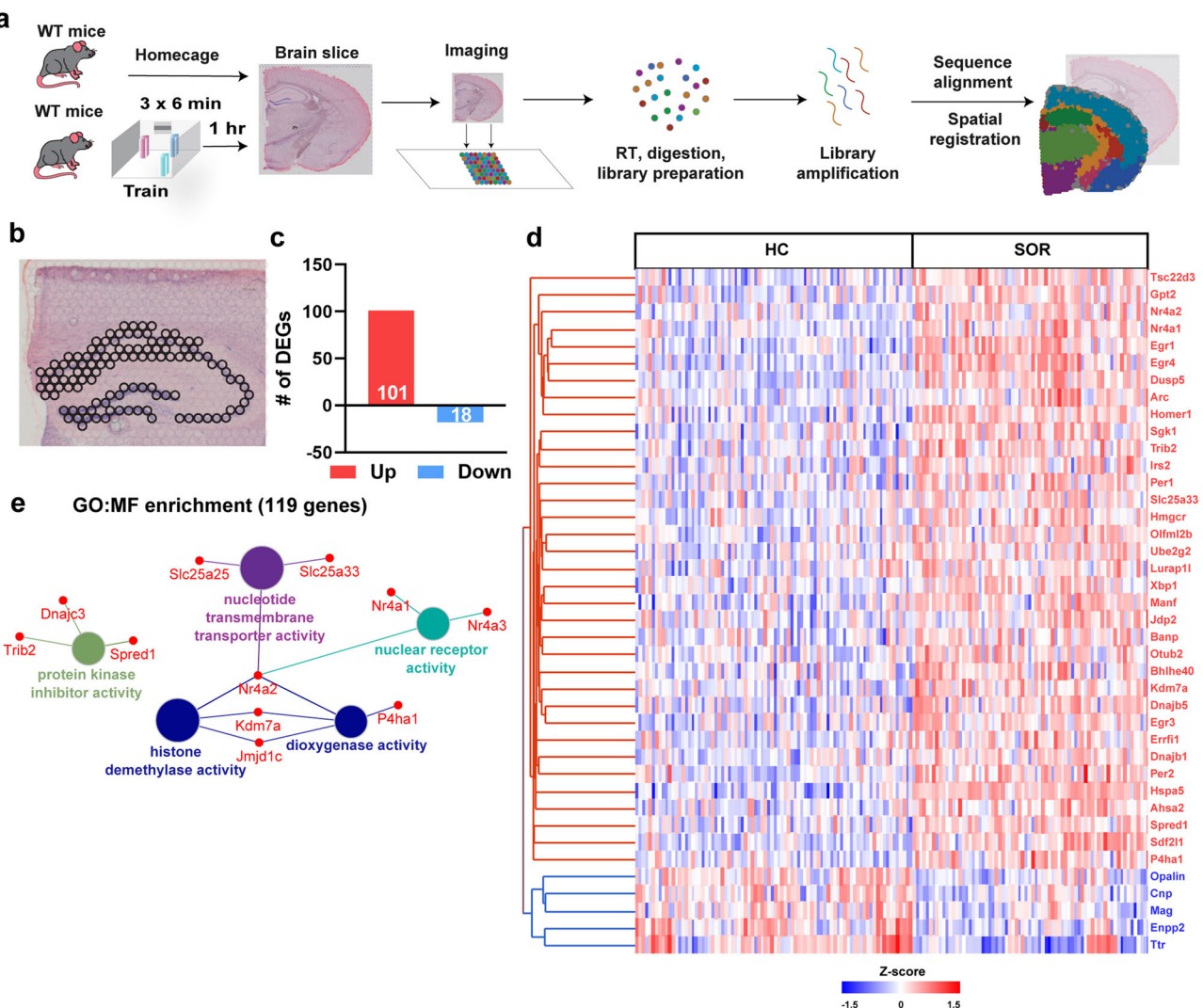

**Fig. 1 | Pseudobulk RNA-seq analysis of spatial transcriptomic data defines learning-induced gene expression in the hippocampus. a** Schematic of the spatial learning paradigm, followed by a graphic description of the Visium pipeline. **b** Visual depiction of spots across all the hippocampal subregions used for pseudobulk RNA-seq analysis. **c** Bar graph illustrating the total number of upregulated and downregulated genes computed from the pseudobulk RNA-seq data. **d** Heat map generated from individual Visium spots of the 40 top significant differentially expressed genes after learning. Red: upregulated, and blue: downregulation genes. **e** Gene Ontology (GO) enrichment analysis performed on all the differentially expressed genes based on their molecular function (MF). Total 7 mice/group, males only.

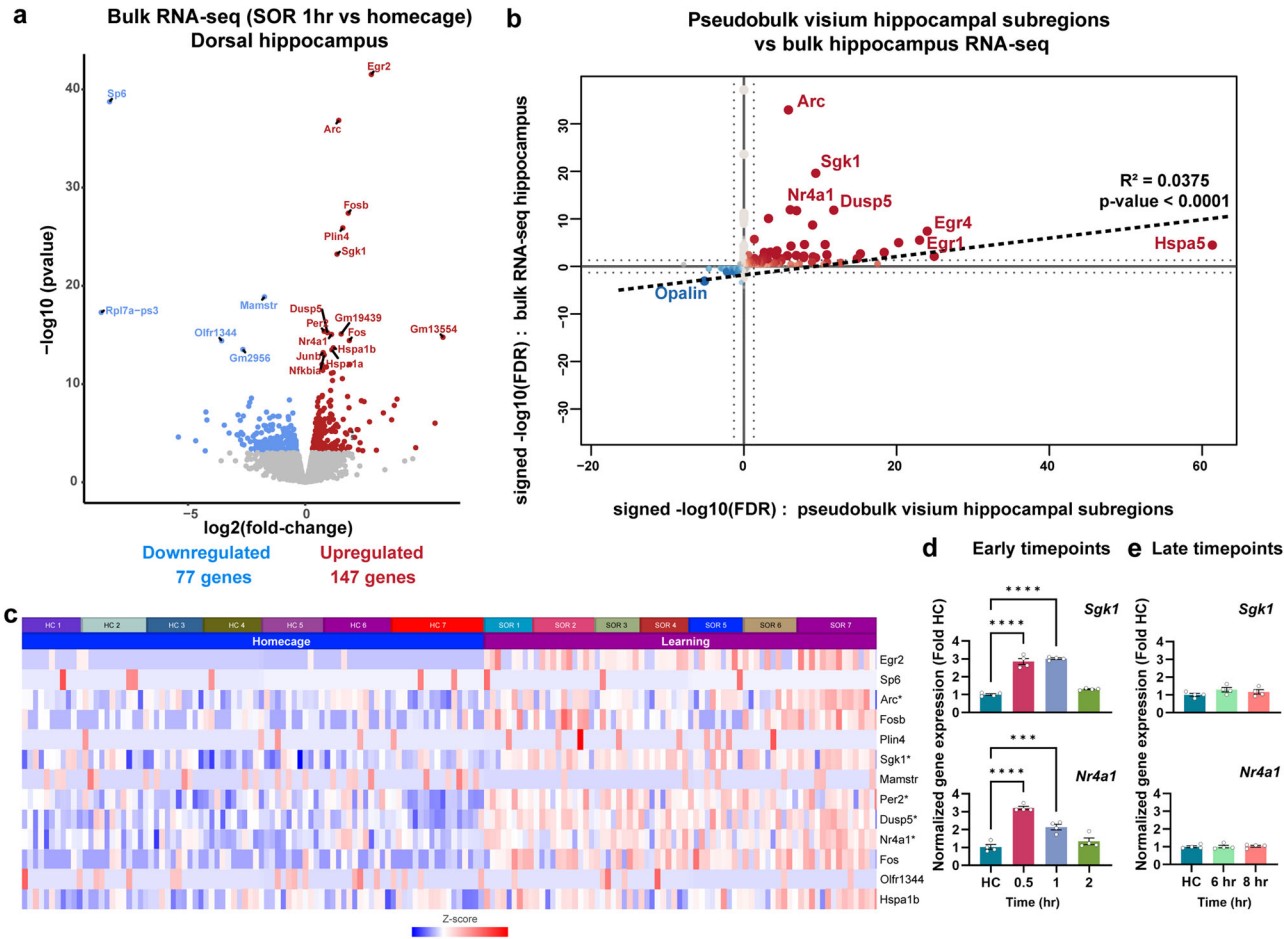

**Fig. 2 | Comparison of the pseudobulk RNA-seq with the bulk RNA-seq dataset after learning. a** Volcano plot illustrating the most significant differentially expressed genes after learning from a bulk RNA-seq experiment performed from the dorsal hippocampus 1 h after learning. homecage (n = 4), SOR (n = 4), males only. **b** Quadrant plot depicting the correlation between differentially expressed genes identified in bulk RNA-seq and pseudobulk RNA-seq. **c** Heatmap showing expression of the top 15 genes from bulk RNA-seq in the pseudobulk samples. *Star indicates significant in both bulk- and pseudobulk-RNA-seq. *Rpl7a-ps3, Gm19439, Olfr1344* genes were not detected in pseudobulk analysis. **d** Expression of *Sgk1* and *Nr4a1* in C57BL/6 J male mice trained in SOR and euthanized at the early timepoints after training (0.5 hour, 1 hour and 2 hours: n = 4 per group), expressed as fold difference from that of mice handled only in the homecage (HC) (baseline controls, n = 4). One-way ANOVA: *Sgk1*: F (3, 12) = 128.3, P < 0.0001. Dunnett's multiple

comparisons tests: ****P < 0.0001 (HC versus 0.5-hour, P = 0.000000038587), ****P < 0.0001 (HC versus 1 hour, P = 0.000000026741), P = 0.0972 (HC versus 2 hour). One-way ANOVA: *Nr4a1*: F (3, 12) = 43.31, P < 0.0001. Dunnett's multiple comparisons tests: ****P < 0.0001 (HC versus 0.5-hour, P = 0.00000064817), ***P = 0.0005 (HC versus 1 hour, P = 0.000495606787944), P = 0.331 (HC versus 2 hour). Error bars represent ± SEM. **e** Expression of *Sgk1* and *Nr4a1* in C57BL/6 J male mice trained in SOR and euthanized at the late timepoints after training (6 hours and 8 hours: n = 4 per group), expressed as fold difference from that of mice handled only in the homecage (HC) (baseline controls, n = 4). One-way ANOVA: *Sgk1*: F (2, 9) = 0.4798, P = 0.6339. Dunnett's multiple comparisons tests: P = 0.1623 (HC versus 6-hour), P = 0.488 (HC versus 8-hour). One-way ANOVA: *Nr4a1*: F (2,9) = 0.083, P = 0.921. Dunnett's multiple comparisons tests: P = 0.9931 (HC versus 6-hour), P = 0.8969 (HC versus 8-hour). Error bars represent ± SEM.

layer, we also investigated the transcriptomic signature exhibited by the less-defined subregions CA1 stratum radiatum and stratum oriens that lack subregion-specific marker genes. CA1 stratum radiatum is the suprapyramidal region containing apical dendrites of pyramidal cells where CA3 to CA1 SC connections are located. CA1 stratum oriens is the infrapyramidal region containing basal dendrites of pyramidal cells where some CA3 to CA1 SC connections are located. However, heterogenous population of interneurons and other non-neuronal cells, such as oligodendrocytes[66], are also scattered through these layers. Differential gene expression analysis from these CA1 regions identified 10 upregulated and 1 downregulated gene in stratum radiatum and 9 upregulated and 9 downregulated genes in stratum oriens (Fig. 3c).

Enrichment network analysis was used to identify the pathways most represented among the DEGs in each hippocampal subregion. The pathways enriched in the CA1 pyramidal layer include nuclear receptor activity and MAP kinase tyrosine/serine/threonine

phosphatase activity (Fig. 3d). In contrast, the pathways in DG include protein kinase inhibitor activity and protein disulfide isomerase activity (Fig. 3e). We utilized an upset plot and a heatmap to compare the differentially expressed genes from each hippocampal subregion (Fig. 3f, g). This analysis identified 51 genes that were exclusively upregulated in DG, 22 genes exclusively upregulated in the CA1 pyramidal layer, and 11 genes upregulated in both CA1 and DG, but not in other hippocampal subregions (Fig. 3f). Some of these 11 common genes are involved in protein folding (*Xbp1, Sdf2l1, Dnajb1*) and the MAPK pathway (*Spred1*). Genes related to activity-driven transcription regulation and MAPK pathway regulation (*Arc, Nr4a2, Per1,* and *Dusp5*) were upregulated both in CA1 and CA2 + CA3 pyramidal layers, while *Nr4a1* and *Egr3* were upregulated in the CA1 pyramidal layer, stratum radiatum and stratum oriens. These findings suggest large-scale transcriptional changes in DG, while CA1 and CA2 + CA3 pyramidal regions showed increased activation state of IEGs linked to engram ensembles following spatial learning.

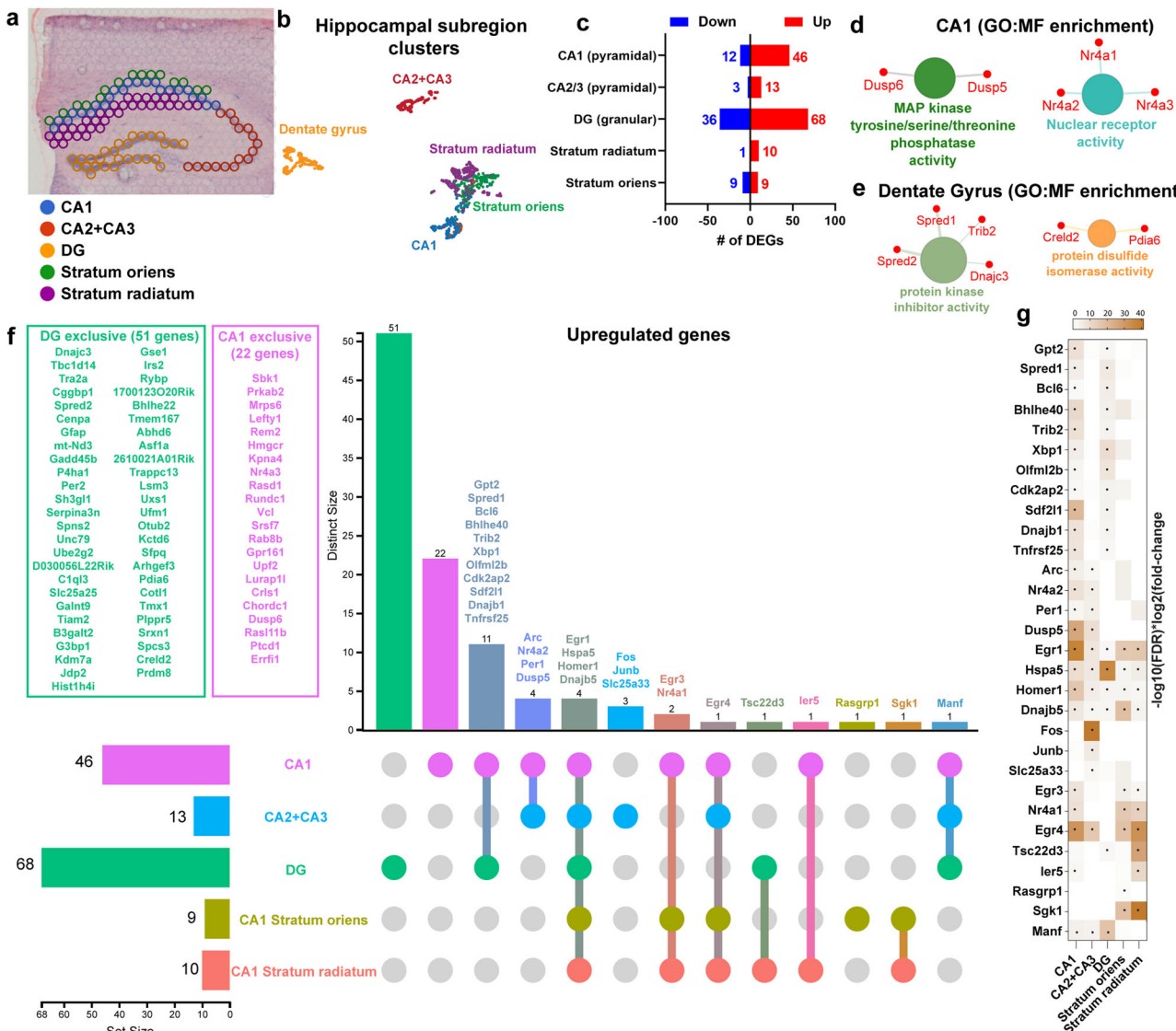

**Fig. 3 | Utilizing spatial transcriptomics to dissect subregion-specific transcriptomic signature of learning in the hippocampus. a** Representative depiction of the Visium spots considered to distinguish hippocampal subregions. **b** UMAP plot showing spot-clusters demarcating the most prominent hippocampal subregions. Homecage (*n* = 7), SOR (*n* = 7). **c** Bar graph depicting the total number of differentially expressed genes corresponding to hippocampal subregions. **d** Gene Ontology (GO) enrichment analysis performed on the differentially upregulated genes in area CA1 pyramidal layer. **e** Gene Ontology (GO) enrichment analysis of all differentially upregulated genes in Dentate Gyrus (DG). **f** UpSet plot illustrating the spatial pattern of all the significantly upregulated learning-induced genes throughout the hippocampus. **g** Heatmap showing extent of expression of genes that are induced in two or more hippocampal subregions. Dots inside the box indicate the genes that pass the threshold of FDR < 0.05 and log2 fold-change >1.4.

Although fewer genes were downregulated following learning compared to upregulated genes, *Kcna4, Usp2*, and *Shisa4* were downregulated in both CA1 and DG subregions (Supplementary Fig. 4). *Kcna4* (Potassium Voltage-Gated Channel Subfamily A Member 4) expression was found to be increased in amyloid beta-induced cognitive impairment[67], suggesting its downregulation could have a role in learning and memory. Similarly, genes encoding two evolutionarily conserved RNA-binding proteins, *Rbm3* and *Cirbp*, were exclusively downregulated in DG (Supplementary Fig. 4) and shown to be differentially expressed in the hippocampus when memory is impaired[68,69]. Among the genes downregulated exclusively in CA1 stratum oriens, *Mbp, Mobp* and *Plp1* are associated with structural constituents of the myelin sheath, and *Opalin* is involved in oligodendrocyte differentiation. While adult oligodendrogenesis and myelination in the cortex are required for memory consolidation[70], the underlying relevance of the downregulation of these genes in hippocampal subregion CA1 stratum oriens is not clear.

## Functional relevance of subregion-specific expression of genes encoding the Nr4a family members

The nuclear receptor 4a (Nr4a) subfamily of transcription factors are critical mediators of memory consolidation. They are robustly upregulated in the hippocampus within minutes after learning to regulate downstream gene expression[56,71,72]. We have previously generated a dominant negative mouse model of Nr4a transcription factors, which expresses a mutant form of Nr4a1 (Nr4ADN) lacking a key transcriptional activation domain[59] and blocking downstream gene expression of all the Nr4a subfamily members through dimerization[73]. Our spatial transcriptomics data revealed upregulation of all the three members of the Nr4a subfamily (*Nr4a1, Nr4a2* and *Nr4a3*) in the CA1 pyramidal layer following learning (Fig. 3). This signature was absent in the dentate gyrus. We validated *Nr4a1* and *Nr4a2* gene expression following SOR training using an in situ-based approach. We found that both *Nr4a1* was upregulated at both 30 min and 1 hr, and *Nr4a2* was upregulated at 1 hr in CA1 but not in DG after learning (Supplementary

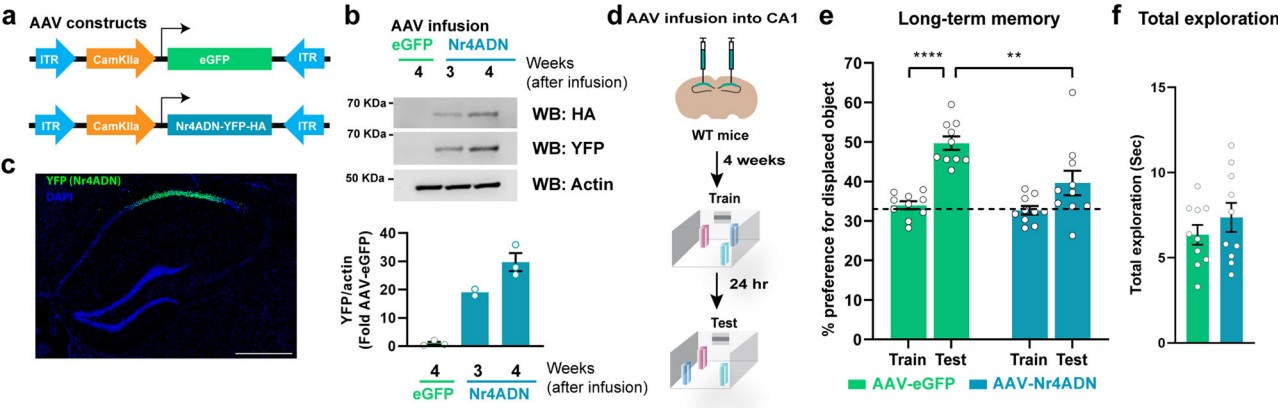

**Fig. 4 | Functional relevance of spatially reserved signatures of learning induced Nr4a family gene expression. a** Design of the constructs packaged into Adeno-associated viruses (AAV) to ectopically express the dominant negative (DN) mutant of Nr4a and EGFP in the CA1 hippocampal subregion. **b** Western Blot analysis showing the time course of viral expression at 3-weeks and 4-weeks after viral infusion. One-way Anova: Šídák's multiple comparisons test: eGFP vs Nr4ADN; $n = 2$ for the 3-weeks Nr4ADN group; $n = 3$ for the eGFP and 4-weeks Nr4ADN groups; males only **c** Immunohistochemistry against YFP to detect the localization and spread of the AAV in the dorsal hippocampus. Scale bar represents 500 µm. **d** Experimental timeline of AAV-infusion into CA1 excitatory neurons followed by spatial learning paradigm. **e** Long-term memory assessment by evaluating preference for the displaced object (DO) in a spatial object recognition (SOR) task. 2-way Anova: Significant sessions (Train-Test) x virus (Nr4ADN-eGFP) interaction: $F_{(1, 18)} = 4.537$, $p = 0.0472$, main effect of sessions: $F_{(1, 18)} = 29.93$, $p < 0.0001$ and main effect of virus: $F_{(1, 18)} = 10.26$, $p = 0.0049$. Šídák's multiple comparisons test: eGFP: train vs test: $p < 0.0001$ ($p = 0.00000139139405$), eGFP (test) vs Nr4ADN (Test): $p = 0.0014$. Nr4ADN ($n = 10$) and eGFP ($n = 10$), males only. Error bars represent ± SEM. **f** Total exploration time of all the objects during SOR for both the experimental groups. Unpaired t test: $t_{(18)} = 2.091$, $p = 0.0510$. Nr4ADN ($n = 10$) and eGFP ($n = 10$), males only. All the bar and dot plots are mean ± SEM.

Fig. 5a–f, and Supplementary Fig. 6). Importantly *Nr4a1* and *Nr4a2* upregulation was found in *Arc*-positive cells in CA1 (Supplementary Fig. 5g–j).

Previous reports suggest that selectively knocking down the expression of either *Nr4a1* or *Nr4a2* in CA1 impairs spatial memory[72]. Therefore, we sought to understand whether blocking the transcriptional activation function of all the three Nr4a family members exclusively in CA1 excitatory neurons would impair long-term memory consolidation. We used an adeno-associated viral construct of Nr4ADN (AAV-Nr4ADN; 2/2 stereotype to enable minimum diffusion across different subregions) under a CaMKIIα promoter to restrict expression to only CA1 excitatory neurons (Fig. 4a, b and c). AAV-Nr4ADN or control (AAV-eGFP) was infused into the dorsal CA1 of wild-type mice 4 weeks before SOR training (Fig. 4d). Both AAV-Nr4ADN and control AAV-eGFP mice showed similar performances in open field during habituation (Supplementary Fig. 7a, b). During training, they showed a progressive decrease in exploration towards objects across training sessions, indicating that learning had occurred (Supplementary Fig. 7c). During the 24 h test session, control mice showed a significant increase in preference for the displaced object during the 24 h SOR test session relative to training, while AAV-Nr4ADN mice failed to show a preference for the displaced object (Fig. 4e). This demonstrates the CA1-specific Nr4ADN mice had impairments in long-term memory. Total exploration of the objects during the test session was unchanged and did not affect preference for the displaced object (Fig. 4f).

Next, we performed a similar approach to block the transcriptional activation function of Nr4a family members in DG, a hippocampal region where the Nr4a genes were not induced following learning. We infused AAV-Nr4ADN or control AAV-eGFP in the DG of wild-type mice (Supplementary Fig. 8a) 4 weeks before SOR training (Supplementary Fig. 8b). Due to low diffusion of AAV2/2, the expression of Nr4ADN was mostly restricted within the upper blade of DG. DG upper blade has been previously shown to exhibit behaviorally induced IEG expression[74,75]. Both AAV-Nr4ADN and AAV-eGFP infused mice showed similar performances during habituation (Supplementary Fig. 8c, d) and during the training sessions (Supplementary Fig. 8e). During the 24 h test session, both the groups of mice exhibited intact long-term memory (Supplementary Fig. 8f). Total exploration of

the objects during the test session was unchanged and did not affect preference for the displaced object (Supplementary Fig. 8g). Thus, blocking the Nr4a transcription function exclusively in DG does not impair memory. This finding indicates that CA1 subregion-specific expression of Nr4a members is essential for long-term memory, providing functional relevance for our spatial transcriptomics finding of CA1 expression of the Nr4a subfamily.

### *Sgk1* is induced in oligodendrocytes of CA1 stratum radiatum and oriens following learning

Among the genes induced in stratum radiatum and oriens, protein kinase *Sgk1* was the only upregulated gene appearing in both the stratum radiatum and oriens but not in the CA1 pyramidal layer (Fig. 5a, b). *Tsc22d3* was found to be specifically induced in stratum radiatum, while *Rasgrp1* was exclusively induced in stratum oriens. This precise signature of *Sgk1* induction following learning was further validated using an in situ-based RNAscope approach (Fig. 5c–f). To understand the distinct cell types responsible for upregulation of *Sgk1* following learning, we performed a single-nuclei RNA seq using a droplet capture independent split-pool barcoding approach (Fig. 5g). We identified 15 clusters of cells from the dorsal hippocampus of mice (Fig. 5h). Interestingly, we found that *Sgk1* expression was significantly induced within the oligodendrocyte populations after learning (Fig. 5i). To further confirm this oligodendrocyte-specific induction of *Sgk1* expression, we performed a single nucleus sequencing assay for transposase-accessible chromatin followed by sequencing (snATAC-seq, Fig. 5j). Based on the accessibility of the promoters of marker genes within each cluster, we were able to identify 11 distinct cell types from dorsal hippocampus of mice (Fig. 5k). Similar to the findings from the snRNA-seq, we found a significant increase in the accessibility of the Sgk1 promoter in oligodendrocytes following learning compared to homecage controls (Fig. 5l). The findings from snRNA- and snATAC-seq suggests that the induction of *Sgk1* seen in the CA1 stratum radiatum and oriens is restricted to oligodendrocyte population within these regions. Thus, using spatial transcriptomics combined with single-nuclei multiomics approaches, we can begin to demonstrate the precise signature of genes within less defined

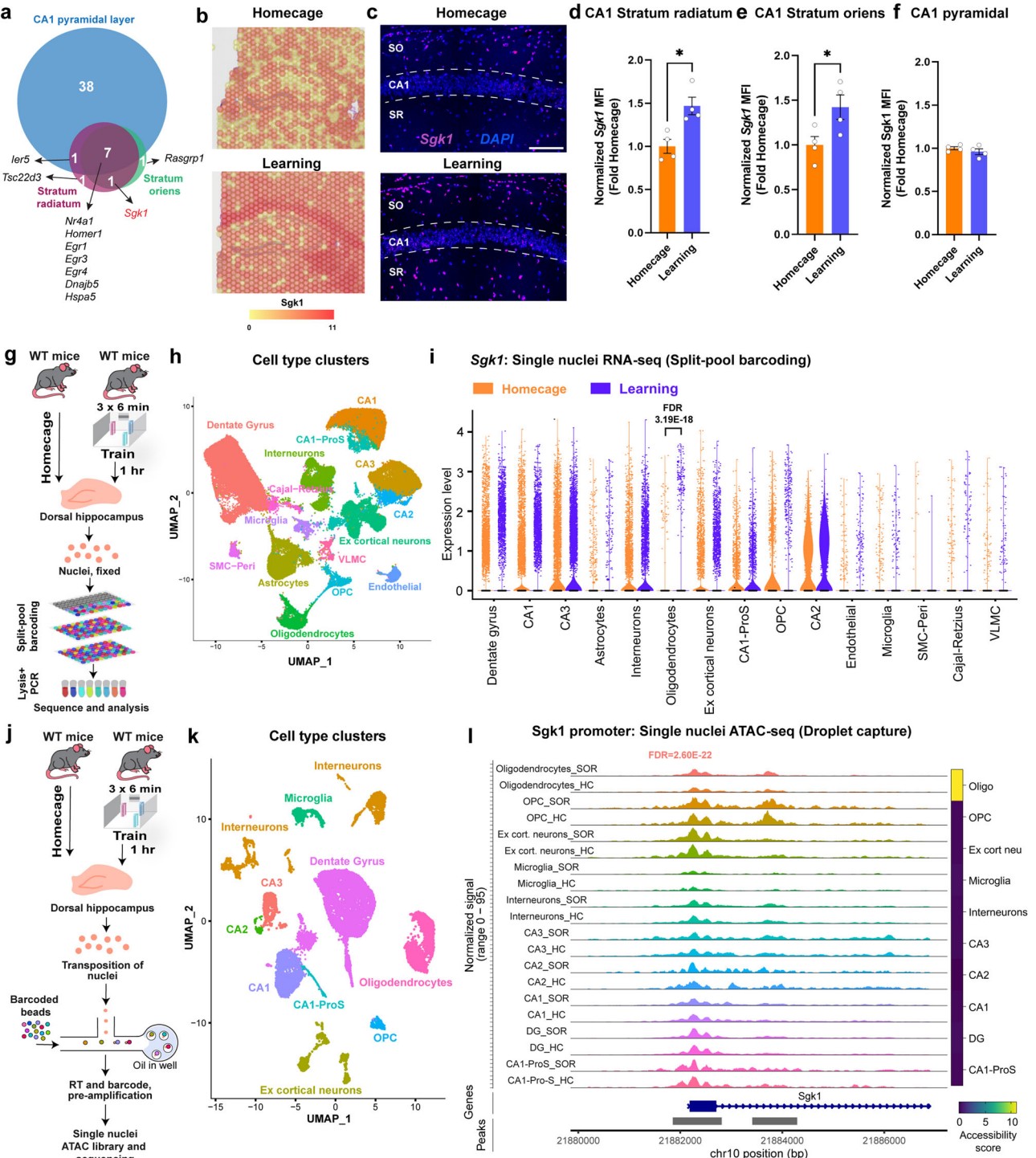

**Fig. 5 | Region and cell type specific expression of Sgk1. a** Venn diagram showing the overlap of upregulated genes exclusive to area CA1 pyramidal layer, Stratum Oriens, and Stratum Radiatum. **b** Representative heatmap of spatial gene expression data showing the expression of *Sgk1*. **c** In situ hybridization showing expression of Sgk1 in CA1 pyramidal later (CA1), CA1 stratum radiatum (SR) and CA1 stratum oriens (SO) in homecaged (HC) and SOR-trained animals. Scale bar represents 200 µm. **d** Quantification of the data from c showing *Sgk1* expression in SR. Homecage (*n* = 4), learning (*n* = 4). Unpaired t test: *t* (6) = 3.613, *p* = 0.0112. Error bars represent ± SEM. **e** Quantification of the data from c showing *Sgk1* expression in SO. Homecage (*n* = 4), learning (*n* = 4). Unpaired t test: *t* (6) = 2.541, *p* = 0.0440. Error bars represent ± SEM. **f** Quantification of the data from c showing *Sgk1* expression in CA1 pyramidal layer. Homecage (*n* = 4), learning (*n* = 4). Unpaired *t* test: t (6) = 1.046, *p* = 0.3359. Error bars represent ± SEM. **g** Experimental design of

single nuclei RNA seq using split-pool barcoding method. Homecage (*n* = 6 mice), SOR (*n* = 6 mice), males only. **h** UMAP from split-pool barcoding approach showing cell type-specific clusters from the dorsal hippocampus. **i** Violin plot showing expression of *Sgk1* across all the cell types between home cage and learning. **j** Experimental design of the single nuclei ATAC seq using droplet capture method. (*n* = 4 mice), SOR (*n* = 4 mice), males only. **k** UMAP showing cell type-specific clusters from dorsal hippocampus based on promoter accessibility of marker genes. **l** Coverage plot showing expression of *Sgk1* across all the cell types between home cage and learning. The quantification of differential accessibility (accessibility score) of the Sgk1 promoter after SOR is represented by the heatmap on the right. The accessibility score was calculated by multiplying fold change with -log10 *p* value.

brain regions that are unable to be distinguished from bulk or single-nuclei transcriptomic datasets alone.

## Discussion

In this study, we uncover a precise transcriptomic signature exhibited by different hippocampal subregions at a critical early timepoint during memory consolidation. While previous work has focused on gene expression changes in the whole hippocampus[38,39,55,76] and individual subregions[40–42,44], our study provides a comprehensive analysis of simultaneous transcriptomic changes spatially distributed across the hippocampal subregions in response to learning. Moreover, we demonstrated that blocking the activity of the Nr4a subfamily of transcription factors selectively within CA1 leads to long-term memory deficits.

The CA1 pyramidal layer, stratum radiatum, and oriens of the dorsal hippocampus are critical regions for encoding spatial memory[77]. While these principal layers play a role in generating spatial maps of the environment[7,46], the granule cells within the DG are thought to provide stable representations of a specific environment[78–80]. In this study, we identified differential expression patterns for important IEGs related to transcriptional regulation in the CA principal layers (CA1 and CA2 + 3) after spatial exploration. *Nr4a1* and *Egr3* were predominantly induced in CA1 subregions, whereas *Arc*, *Nr4a2*, *Per1*, and *Dusp5* were upregulated in CA1 and CA2 + 3 regions. IEGs *Egr1* and *Homer1* were found to be upregulated in all subregions studied, while *Gadd45b* and *Per2* were induced exclusively in DG. This subregion-specific expression could contribute to the afferent and efferent connections within the hippocampus, observed within our specific time point at 1 h. Differential gene induction within a timeframe has been correlated with the activation of engram ensembles[4–7] and place codes underlying spatial maps[7,46]. Although we found that the CA1 subregion exhibited a greater number of IEGs associated with engram ensembles[5–7], we observe a greater number of DEGs in DG compared to CA1 following spatial exploration. This is consistent with single nuclei data from activated and non-activated neurons from DG and CA1[48]. Our study identified transcriptomic signatures within the two understudied hippocampal subregions, CA1 stratum radiatum and oriens, which have been challenging to delineate using conventional single-cell sequencing strategies due to a lack of specific marker genes. Overall, our study elucidates the transcriptomic diversity that exists among hippocampal subregions and helped define a spatial map during an early window of memory consolidation.

The Nr4a subfamily, *Nr4a1, Nr4a2,* and *Nr4a3*, serve as major regulators of gene expression in the hippocampus during memory consolidation[59,60,72,73,81,82]. *Nr4a1* and *Nr4a2* are necessary for object location memory in the dorsal hippocampus, while only *Nr4a2* is necessary for object recognition memory[72]. Impairments in Nr4a function[59,83] lead to long-term memory deficits[56,59] and reduced transcription-dependent long-term potentiation (LTP) in CA1[84]. On the contrary, overexpression or pharmacological activation of Nr4a family members ameliorates memory deficits in mouse models of Alzheimer's disease and related dementias (ADRD) and age-associated memory decline[59,60,71,85]. Our identification of upregulation of all Nr4a subfamily members after learning in CA1 confirms findings from previous studies using hippocampus-dependent learning tasks[71,72,86,87]. We further demonstrate that the CA1-specific expression of the Nr4a subfamily is functionally relevant to long-term spatial memory within the hippocampus. Thus, understanding the spatial component of learning-induced transcriptomic heterogeneity in the hippocampal cell layers strongly supports the concept of subregion-specific dissociation as a molecular mechanism underlying memory consolidation.

The basal dendrites of CA1 pyramidal neurons make up stratum oriens, while stratum radiatum consists of apical dendrites. Both stratum radiatum and oriens receive inputs from CA3 Schaffer collaterals[88]. We found upregulation in *Nr4a1, Homer1, Egr1, Egr3, Egr4,*

*Dnajb5, and Hspa5* in the CA1 pyramidal layer, CA1 stratum radiatum and oriens. Interestingly, *Sgk1* was restricted only to the stratum oriens and stratum radiatum and is known to have a functional role in memory consolidation. Expression of a dominant negative Sgk1 within CA1 impaired spatial memory[32], whereas constitutively active Sgk1 enhanced spatial memory[31]. Furthermore, in an APP/PS1-based ADRD model, *Sgk1* was downregulated in the hippocampus, whereas over-expression of Sgk1 could ameliorate spatial memory deficits[34]. Sgk1 regulates the *Egr1* expression[89], an IEG that we found upregulated in all subregions of the hippocampus following learning. Our single-nuclei transcriptomic and epigenomic studies revealed that *Sgk1* is induced within oligodendrocytes of stratum radiatum and oriens. Oligodendrocytes form myelin sheath for the axons of CA1 pyramidal cells[66], suggesting that the role of Sgk1 in the dorsal hippocampus lies within these supporting subregions of CA1. Studying the spatial patterns of learning-responsive genes like *Sgk1* further defines the role of each hippocampal subregion in memory consolidation.

We identified two upregulated pathways in DG that are involved in protein kinase inhibitor activity and protein processing in the endoplasmic reticulum (ER). We have recently shown that learning induces the expression of molecular chaperones localized in the ER, and this protein folding machinery is critical in synaptic plasticity and long-term memory consolidation[59]. Here, our spatial transcriptomics data shows upregulation of genes encoding chaperones in distinct subregions; *Hspa5* and *Dnajb5* across all the hippocampal subregions, *Xbp1, Sdf2l1* and *Dnajb1* in areas CA1 and DG, and *Pdia6* and *Creld2* exclusively in DG. This suggests that DG could have a prominent role in ER protein processing during an early time point after spatial learning, as ER chaperones are indeed critical mediators of long-term memory storage[59]. This work also suggests that there may be distinct protein processing complexes in different hippocampal subregions to facilitate the folding and trafficking of distinct proteins during memory consolidation.

We compared our DEGs detected from pseudobulk spatial transcriptomics with bulk RNA-seq and found a significant correlation between the profiles. Although we observed an overlap in DEGs, how the libraries were prepared, the amount of tissue and subregions sequenced, and statistical approach for each technique explains why we observe differences in gene expression from these two techniques. We identified 105 more DEGs by bulk RNA-seq, possibly because this approach included all RNA, except ribosomal RNA, within all cell types and subregions of the entire dorsal hippocampus. Pseudobulk spatial transcriptomics included the poly-A mRNAs in barcoded dots selected within principal cell layers, stratum radiatum and oriens on a 10 μm brain slice. Any additional differences in overlap can be attributed to the type of statistical test used to compute significant changes in gene expression. Pseudobulk was analyzed by the rank-sum Kruskal-Wallis test, while bulk RNA-seq was analyzed by the conventional EdgeR statistical test. We could not use the same statistical test because EdgeR utilizes the mean instead of the median to compute fold-change and may increase the number of outliers or false positives for pseudobulk analysis. Therefore, some DEGs in bulk RNA-seq appeared as a lower fold-change in pseudobulk because they did not surpass the 1.4 threshold for the Kruskal-Wallis test, yet they still displayed differential expression following learning.

Our work demonstrates that the subregions of the dorsal hippocampus uniquely respond to learning by exhibiting distinct transcriptomic signatures. These subregions differ by their circuitry, cell types, and electrophysiological features. A criticism of the spatial transcriptomic approach is that it lacks cell-type specific information, yet we see changes in some non-neuronal genes after learning, such as *Sgk1*. Therefore, future studies will need to address heterogeneity between cell types of subregions and how they respond to neuronal activation after learning. Thus, combining spatial transcriptomics with single-cell transcriptomics and high throughput in situ approaches[90]

provides further insights into cell-type specific changes in gene expression across different hippocampal subregions, as well as subregions that are challenging to study due to a lack of marker genes or a small density of cells. This approach could be important for understanding differential gene expression patterns across the ventral hippocampus and other brain regions[91–93] at various timepoints during memory consolidation. Our attempt to elucidate the spatial transcriptomic signature of memory provides the groundwork for future studies to understand the precise gene expression patterns underlying memory consolidation, and whether these signatures are affected in neurological disorders associated with memory impairments.

## Methods

### Data reporting

No statistical methods were used to predetermine the sample size.

### Mouse lines

Adult male C57BL/6 J mice were purchased from Jackson Laboratories were 2–3 months age during behavioral or biochemical experiments. All mice had free access to food and water and were maintained under pathogen-free conditions with a 12 h light/dark cycle, at a temperature of 21–22 °C and a relative humidity of 60–70% in the Animal care facility of the University of Iowa. All experiments were conducted according to US National Institutes of Health guidelines for animal care and use and were approved by the Institutional Animal Care and Use Committee of the University of Iowa, Iowa.

### Adeno-associated virus (AAV) constructs and stereotactic surgeries

$AAV_{2.2}$-CaMKIIα-Nr4ADN and $AAV_{2.2}$-CaMKIIα-EGFP were purchased from VectorBuilder (VectorBuilder Inc). Mice were anesthetized using 5% isoflurane, following which a steady flow of 2.5% isoflurane was maintained throughout the duration of the stereotactic surgery[59]. 1 μl of respective AAVs were injected into the dorsal hippocampal subregion CA1 (coordinates: anteroposterior, −1.9 mm, mediolateral, ±1.5 mm, and 1.5 mm below bregma) or DG (coordinates: anteroposterior, −1.9 mm, mediolateral, ±1.3 mm, and 2 mm below bregma). Following viral infusion, drill holes were closed with bone wax (Lukens) and the incisions were sutured.

### Spatial object recognition (SOR) task

All the behavioral tasks were performed during the light cycle between Zeitgeber time (ZT) 0 to 2. Animals were individually housed for 7 days before training. Animals were handled for 2 mins each day for 5 successive days before training. On the day of training, animals were habituated in an open field for 6 min, followed by three 6-min sessions inside the same open field containing three different glass objects placed in specific spatial locations with respective to a spatial cue inside the arena. An intertrial interval of 5 min was set in-between the three training sessions. 24 h later, the animals were returned to the arena with one of the objects displaced to a novel spatial coordinate. Exploration time around all the objects were then manually scored[59]. Percent preference towards the displaced object was calculated using the following equation:

$$Percent\ preference\ for\ displaced\ object = \frac{(exploration\ towards\ the\ displaced\ object)}{(total\ exploration\ towards\ all\ objects)} x\ 100$$

For spatial gene expression, single nuclei experiments, and in situ experiments, animals were euthanized by cervical dislocation 1 h after the last training trial in the SOR task. Whole brains or hippocampal tissue were flash frozen and stored at −80 °C. Homecage animals were also euthanized at the same time to eliminate any circadian effect. Quantification of time spent in the inner and outer zones in the open field during the habituation sessions was performed using the Etho-VisionXT software (Noldus Instruments)[60].

### Spatial transcriptomics sample preparation

After rapidly euthanized by cervical dislocation, the brains from 8 mice were rapidly extracted and flash- frozen with −70 °C isopentane for 5 min. Frozen brains were stored at −80 C until sectioning. Mouse-frozen brains were embedded in an optimal cutting temperature medium (OCT) and cryosectioned at −20 °C with the Leica CM3050 S Cryostat. 10-microns of coronal sections from the brain region with dorsal hippocampus were placed on chilled Visium Tissue Optimization Slides (10X Genomics) and Visium Spatial Gene Expression Slides (10X Genomics). We used 8 brains (4 homecage and 4 SOR-trained animals), 1 slice per animal were placed on barcoded frames on a Visium slide (10X Genomics). Thus, we used 2 Visium slides with 4 frames each. Visium slides with the sections were fixed, stained, and imaged with Hematoxylin and Eosin using a 20X objective on an Olympus BX61 Upright Microscope. Tissue was then permeabilized for 18 min, which was established an optimal permeabilization time based on tissue optimization time-course experiments. The poly-A mRNAs from the slices were released and captured by the poly(dT) primers and pre-coated on the slide, including a spatial barcode and a Unique Molecular Identifiers (UMIs). After reverse transcription and second-strand synthesis, the amplified cDNA samples from the Visium slides were transferred, purified, and quantified for library preparation. The fragmented cDNA samples were used to construct sequencing for Visium spatial transcriptome on a NovaSeq 6000 (Illumina) at a sequencing depth of 150 million total read pairs per mouse Visium sample.

### Spatial transcriptomics library preparation and sequencing

Sequencing libraries were prepared by the Iowa Institute of Human Genetics (IIHG) Genomics Division, according to the Visium Spatial Gene Expression User Guide. Each pooled library was sequenced on an Illumina NovaSeq 6000 using SBS chemistry v1.5 for 100 cycles, at a sequencing depth of 200 million total read pairs. Data processing of Visium data, raw FASTQ files and images were output with Space Ranger software (Version 1.3.1) and analyzed downstream by Partek Flow (Partek Inc.) with their single-cell analysis pipeline, mm10 reference genome was used for gene alignment.

### Spatial transcriptomics data analysis

Space Ranger outputs were uploaded to Partek Flow (Build version 10.0.23.0720) for downstream analyses. The read counts were normalized by the counts per million (CPM) method and transformed to log2(CPM + 1). A general linear model was applied to correct for batch effect between the two sets of experiment. Hippocampal subregions were selected based on biological knowledge using anatomical structures apparent on the H&E staining images. The pyramidal layers of CA1, CA2 + CA3 and the granular and molecular layer of DG were selected for their role in neuronal excitability, synaptic plasticity, and memory. Additionally, CA1 stratum radiatum and oriens were also selected due to their roles in neuronal circuitry. Differential gene expression analysis was performed using the non-parametric Kruskal-Wallis rank sum test because this type of tests have been the most widely used approach in the field of single-cell transcriptomics (Squair et al. 2021). Because each cell is assumed to be a biological replicate in scRNA-seq, the same assumption is made here for each visium spot, which generates a big sample size that is handled correctly by the Kruskal-Wallis test. Gene-specific analyses were filtered with false discovery rate (FDR) < 0.05 and fold change > |1.4 | . The spots assigned to the hippocampus of a Visium slide are spaced apart from each other (i.e., do not overlap) and correspond to a unique set of cells within the tissue sample. Therefore, the expression profile measured at each spot can be treated as an independent observation. This approach gives a sufficient sample size for statistical power to detect differences in gene

expression[94]. Statistical analyses of spots as sample sizes must account for spatial autocorrelation of gene expression levels and genes with statistically significant spatial patterns[95]. Spatial autocorrelation is a statistical phenomenon that occurs when observations within a spatially defined region are more similar to each other than to observations from other regions. For our spatial transcriptomics, gene expression levels measured at adjacent spots on the Visium slide are likely to be more similar to each other than to spots that are further apart. By accounting for spatial autocorrelation, we can depict each spot as an independent observation for accurate and precise transcriptomic analysis.

### Bulk RNA extraction, cDNA preparation and gene expression analysis

Dorsal hippocampi were dissected and immediately stored at −80 °C in RNAlater solution (Ambion). For RNA total extraction, hippocampi were homogenized in Qiazol (Qiagen) using stainless steel beads (Qiagen). Chloroform was then added, and the homogenate was centrifuged at $12,000 \times g$ at RT for 15 min. Aqueous phase containing RNA was precipitated using ethanol and then cleaned using the RNeasy kit (Qiagen). RNA was eluted in nuclease-free water, treated with DNase (Qiagen) at RT for 25 min and precipitated in ethanol, sodium acetate (pH 5.2) and glycogen overnight at −20 °C. Precipitated RNA samples were centrifuged at top speed at RT for 20 min, washed with 70% ethanol and centrifuged at top speed for 5 min, dried and resuspended in nuclease-free water. RNA concentrations were estimated using a Nanodrop (Thermo Fisher Scientific). cDNAs were prepared from 1 μg RNA using the SuperScript™ IV First-Strand Synthesis System (Ambion). Real-time RT-PCR reactions were performed in a 384-well optical reaction plate with optical adhesive covers (Life Technologies). Each reaction was composed of 2.25 μl cDNA (2 ng/ul), 2.5 μl Fast SYBR™ Green Master Mix (Thermo Fisher Scientific), and 0.25 μl of primer mix (IDT). Three technical replicates per reaction was performed on the QuantStudio 7 Flex Real-Time PCR system (Applied Biosystems, Life Technologies). Data was normalized to housekeeping genes (*Tubulin*, *Pgk1* and *Hprt*) and $2^{(-\Delta\Delta Ct)}$ method was used for gene expression analysis. The oligonucleotide sequence of the qRT-PCR primers used were: Sgk1 (FW: 5′ GCCAAGTCCCTCTCAACAAA 3′; Rev 5′ CCCTTTCCGAT-CACTTTCAA 3′), Nr4a1 (FW 5′ AAAATCCCTGGCTTCATTGAG 3′; REV 5′ TTTAGATCGGTATGCCAGGCG 3′), Tubulin (FW 5′ ATGCGCGAGTG-CATTTCAG 3′; REV 5′ CACCAATGGTCTTATCGCTGG 3′), Pgk1 (FW 5′ CGAGCCTCACTGTCCAAACT 3′; REV 5′ TCTGTGGCA-GATTCACACCC 3′), Hprt (FW 5′ TTGCTGACCTGCTGGATTACA 3′; REV 5′ CCCCGTTGACTGATCATTACA 3′).

### Library preparation and sequencing from bulk RNA

RNA libraries were prepared at the Iowa Institute of Human Genetics (IIHG), Genomics Division, using the Illumina TruSeq Stranded Total RNA with Ribo-Zero gold sample preparation kit (Illumina, Inc., San Diego, CA). Library concentrations were measured using KAPA Illumina Library Quantification Kit (KAPA Biosystems, Wilmington, MA). Pooled libraries were sequenced on Illumina NovaSeq6000 sequencer with 150-bp Paired-End chemistry (Illumina) at the IIHG core.

### Bulk RNA-seq analysis

Sequencing data was processed with the bcbio-nextgen pipeline (https://github.com/bcbio/bcbio-nextgen). The pipeline uses STAR[96] to align reads to the genome and quantifies expression at the gene level with featureCounts[97]. All further analyses were performed using R. For gene level count data, the R package EDASeq was used to account for sequencing depth (upper quartile normalization)[98]. Latent sources of variation in expression levels were assessed and accounted for using RUVSeq (RUVs)[99]. Appropriate choice of the RUVSeq parameter k was determined through inspection of RLE plots and PCA plots. Differential expression analysis was conducted using edgeR[100].

### Single nuclei RNA-sequencing using SPLiT-pool barcoding

Frozen hippocampal tissue was homogenized in 700 μl of homogenization buffer (250 mM Sucrose, 25 mM KCl, 5 mM MgCl₂, 10 mM Tris buffer (pH-8), 1 μM DTT, and 0.1% Triton X-100; supplemented with RNAse Inhibitors) using Dounce homogenizers. The homogenate was filtered through 40 μm strainer and centrifuged at 600*g for 4 min at 4 °C. The pellet was resuspended in 1 ml of 1X PBS containing 1% BSA and centrifuged at 600*g for 4 minutes at 4 °C. The pellet was resuspended in 200 μl of 1X PBS and passed through a 40 μm cell strainer.

Postnuclear extraction, nuclear fixation was performed according to the manufacturer's manual (Fixation Kit, Parse Biosciences). A total of 4 million nuclei was transferred into a 15 mL falcon tube, centrifuged at 200*g for 10 min at 4 °C, and resuspended in ice-cold Nuclei Buffer containing 0.75% BSA. Resuspended nuclei were then strained through a 40 μm cell strainer, after which 250 μl of Nuclei Fixation Solution was added. After 10 minutes of incubation on ice, 80 μl od ice-cold Nuclei Permeabilization Buffer was added, and the nuclei were then centrifuged at 200*g for 10 min at 4 °C. The nuclear pellet was resuspended in 300 μl of Nuclei Buffer and passed through a 40 μm cell strainer. Nuclei DMSO was then added to the nuclei and stored in −80 °C until the commencement of single-nuclei RNA-Seq.

Single nuclei RNA-Seq was also performed according to the manufacturer's protocol (Single Whole Cell Transcriptome Kit, Parse Biosciences). Briefly, RNA from the fixed nuclei were subjected to multiple rounds of reverse transcription barcoding and ligation barcoding in customized 96-well thermocycler plates that contain barcoded primers in each well. After the final round of thermocycling, nuclei were pooled together and distributed into 8 sublibraries. Barcoded DNA in each sublibrary was then amplified, following which Template Switching and SPRI Clean-Up steps were performed. Sublibraries were then subjected to Fragmentation, End Repair, and A-Tailing; followed by Post-Fragmentation Double sided SPRI selection, Adaptor Ligation, and Sublibrary Index PCR. Finally, post-Amplification double-sided size selection was performed before sequencing them on an Illumina Novaseq Sequencer.

### SPLiT-seq analysis

Prior to running the pipeline, Fastq files of each sublibrary from separate lanes were concatenated into one (i.e. cat sublibrary1_read1_L001.fastq.gz sublibrary1_read1_L002.fastq.gz > sublibrary1_read1_concat.fastq.gz). Pre-processing of SPLIT-seq data was performed using the split-pipe pipeline developed by Parse Bioscience (v0.9.6p), --mode comb --kit WT. First the data were processed by the pipeline from each sublibrary individually (using split-pipe --mode all). Then the processed data from each sublibrary were combined into a single dataset (using split-pipe --mode comb).

Downstream analyses were carried out using R programming language (version 4.2.2) and packages like Seurat. The raw count matrix was obtained by reading and parsing the pipeline output files using the "ReadParseBio" function. Empty gene names in the matrix were identified and replaced with the label "unknown". Cell metadata, including sample information, was read from a CSV file. A Seurat object was created, specifying minimum gene and cell thresholds for filtering, and the sample categories were assigned based on the original "sample" column in the metadata. Normalization of the expression data was performed using the NormalizeData function in Seurat. The most 2000 variable features were identified using the FindVariableFeatures function with the "vst" selection method. The data were then scaled using the ScaleData function. Principal component analysis (PCA) was conducted using the RunPCA function to reduce the dimensionality of the data (npcs = 50). The resulting principal components were used to find cell-to-cell relationships by calculating neighbors and clusters. The FindNeighbors function identified neighboring cells based on PCA dimensions (reduction = "pca", dims = 1:20), and the FindClusters function assigned cells to clusters using a 0.3 resolution. Cluster

identities were assigned to cells using the Idents function. Clusters were reordered according to their similarity using the BuildCluster-Tree function. To visualize the cell clusters, non-linear dimensional reduction technique such as Uniform Manifold Approximation and Projection (UMAP) was applied using the RunUMAP function (reduction = "pca", dims = 1:20), and the DimPlot function was used to create a UMAP plot with clusters color-coded (reduction = "umap"). To assess the expression of specific genes in certain clusters, a list of marker genes was generated. The FeaturePlot function was used to visualize the expression of these marker genes. Additionally, a list of clusters was created to group related cell types for subsequent merging. Cells within clusters were merged based on the created list of cell types and corresponding clusters using a for loop. The merged clusters were assigned new identities. Differential gene expression analysis was performed using the FindMarkers function with the Wilcoxon statistical test, comparing samples from the HC and SOR conditions. Genes that surpass the 0.2 threshold log2FC and FDR < 0.05 were considered significant.

## Nuclei Isolation and single nuclei ATAC-Seq

Nuclei isolation for Single-Cell ATAC-Seq was performed according to the manufacturer's manual (Chromium Nuclei Isolation Kit, 10x Genomics). Briefly, frozen hippocampal tissue was homogenized in 500 µl of pre-chilled lysis buffer using Dounce homogenizers. The homogenate was then transferred into pre-chilled Nuclei Isolation Columns, and the columns were centrifuged at 16000*g for 30 seconds at 4 °C. The flowthrough was briefly vortexed for 10 seconds, followed by centrifugation at 500*g for 3 min at 4 °C. The nuclear pellet was resuspended in 500 µl of Debris Removal Buffer and centrifuged at 700*g for 10 min at 4 °C. The nuclear pellet was then resuspended in 1 ml of Wash Buffer, centrifuged at 500*g for 5 min at 4 °C, and finally resuspended in 50 µl of Resuspension Buffer. The Single Cell ATAC-Seq protocol, comprising of the Transposition, GEM Generation, Barcoding, and Library Construction steps were carried out according to the manufacturer's protocol (Chromium Next GEM Single Cell ATAC Kit, 10x Genomics). Libraries were sequenced in paired end in Illumina Novaseq.

## Single nuclei ATAC-sequencing data analysis

scATAC-seq data analysis was performed using the R programming language (version 4.2.2) and packages like Seurat (version 4.3.0) and Signac (version 1.9.0). The analysis pipeline assumed that the cell ranger-ATAC pipeline (cellranger-atac-2.0.0) from 10x Genomics had already been run.

Seurat objects were created for each individual dataset (HC and SOR), using the peak/cell matrix and cell metadata files were generated by cellranger-atac. Fragment files were associated with each Seurat object. Quality control metrics were computed for each cell, including nucleosome signal score, TSS enrichment score, blacklist ratio, and fraction of reads in peaks. Outlier cells were removed based on predefined criteria (peak_region_fragments >3000 & peak_region_fragments <20000 & pct_reads_in_peaks >15 & nucleosome_signal <4 & TSS.enrichment >2). Gene annotations for the mouse genome were added to the Seurat object using the Get-GRangesFromEnsDb function. Normalization and linear dimensional reduction were performed using the Seurat functions FindTopFeatures, RunTFIDF, and RunSVD. Non-linear dimension reduction was performed using the RunUMAP function (reduction = 'lsi', dims = 2:30). Clustering was performed using the FindNeighbors (reduction = 'lsi', dims = 2:30) and FindClusters (algorithm = 3, resolution = 1.2) functions. All samples were merged into one Seurat object using the Merge function. The combined data was visualized using UMAP to ensure that no cluster of cells were present in one condition and not the other. Gene activity matrix was created with a window of 2000 bp before and after the transcription start site of each gene and added as a new assay to the Seurat object, and the data were normalized using the NormalizeData function (assay = 'RNA', normalization.method = 'LogNormalize', scale.factor = median(seuratobject$nCount_RNA)). Identification of cell types was performed by loading previously processed SPLIT-seq data for homecage condition. Transfer anchors were then identified between the SPLIT-seq data and the gene activity matrix of the scATAC-seq data using the FindTransferAnchors function. Predicted labels from the SPLIT-seq data were transferred to the scATAC-seq data using the TransferData function. Finally, differential accessibility analysis was performed on the gene activity matrix to identify differentially accessible promoters between conditions for a specific cell type. Coverage plots were generated for specific genes of interest.

## Molecular function enrichment analysis

The identified DEGs were analyzed for molecular function enrichment analysis by using the ClueGO and CluePedia plug-ins of the Cytoscape 3.9.0 software in "Functional analysis" mode against the Gene Ontology Molecular Function (4691 terms) database. The GO Term Fusion was used allowing for the fusion of GO parent-child terms based on similar associated genes. The GO Term Connectivity had a kappa score of 0.4. The enrichment was performed using a two-sided hypergeometric test. The $p$ values were corrected with a Bonferroni step-down approach. Only significant molecular function with corrected $p$ values < 0.05 were displayed. UpSet plots were generated using the online software ExpressAnalyst. Data was plotted using the distinct mode.

## In situ hybridization (RNAscope)

In situ hybridization was performed on 20 µm coronal brain sections from SOR-trained and homecaged animals using the RNAscope™ Multiplex Fluorescent Reagent Kit v2 (Advanced Cell Diagnostics). Briefly, fixed frozen brains were sectioned on a Cryostat and mounted on Superfrost™ Plus microscope slides (Fisherbrand). Slides then underwent serial dehydration in Ethanol, followed by Hydrogen Peroxide treatment, Target Retrieval, and Protease III treatment. Hybridization of probes was done at 40 °C for 2 h in an HybEZ oven using probes against *Sgk1, Nr4a1, Nr4a2,* and *Arc.* The probe signals were amplified with Preamplifier 1 (Amp 1-FL) and counterstained with OPAL dyes (Akoya Biosciences). The slides were mounted with Vectashield Antifade Mounting Medium with DAPI (Vector Laboratories) and stored in 4 °C until they were imaged.

## Confocal imaging and RNAScope analysis

Following in situ hybridization, high magnification images of the hippocampus were obtained in an Olympus FV 3000 confocal microscope using a 40X NA = 1.25 oil immersion objective at 800 × 800-pixel resolution and 1.5X optical zoom. All images (16 bit) were acquired with identical settings for laser power, detector gain and pinhole diameter for each experiment and between experiments. Images from the different channels were stacked and projected at maximum intensity using ImageJ (NIH). Hippocampal subregion-specific Mean Fluorescence Intensity (MFI) and colocalization analyses were performed using ImageJ plugins.

## Western blot analysis

Whole-cell lysates were run on a 4–20% Tris-HCl Protein Gel (BIO-RAD, # 3450033) and transferred to methanol-activated polyvinylidene difluoride membranes. Membranes were blocked with Odyssey® Blocking Buffer in TBS (LI-COR) and incubated overnight at 4 °C with the following primary antibodies: pan-HA (1:1000, Cell signaling), YFP (1:1000, Abcam), and Actin (1:10,000, ThermoFisher Scientific). Membranes were washed and incubated with anti-rabbit IRDye 800LT (1:5000, LI-COR, # 926-32211) and anti-mouse IRDye 680CW (1:5000,

LI-COR, #926-68022). Images were acquired using the Odyssey Infrared Imaging System (LI-COR). Quantification of western blot bands was performed using Image Studio Lite ver5.2 (LI-COR).

## Immunohistochemistry and confocal imaging
Animals were anesthetized using a steady flow of 5% isoflurane and perfused with 4% PFA, and 20 μm coronal brain sections were made in a cryostat. Free-floating sections were washed with PBS and mounted on on Superfrost™ Plus microscope slides (Fisherbrand). The sections were air-dried, followed by coverslip mounting with Vectashield® Antifade Mounting Medium with DAPI (Vector Laboratories). Slides were then imaged using the Olympus FV3000 confocal microscope with a 10X NA = 0.4 objective at 800 × 800-pixel resolution.

## Statistics
Behavioral and biochemical data were analyzed using unpaired two-tailed $t$ tests and either one-way or two-way ANOVAs (in some cases with repeated measures as the within subject variable). Sidak's or Dunnett's multiple comparison tests were used for post hoc analyses where needed. Differences were considered statistically significant when $p < 0.05$. As indicated for each figure panel, all data are plotted in either bar graphs, in which symbols represent each data point, or in dot plots, where each symbol represents an individual data point. Graphs were plotted as mean ± SEM.

## Ethics
All procedures on mice in this study were approved by the Institutional Care and Use Committee at the University of Iowa.

## Reporting summary
Further information on research design is available in the Nature Portfolio Reporting Summary linked to this article.

## Data availability
The sequencing data generated in this study have been deposited in the NCBI Gene Expression Omnibus (GEO) database under accession code GSE223066. Sequencing files for bulk RNA-seq, spatial transcriptomic, SPLIT-seq and snATAC-seq have been made publicly available through GSE223066.

The mm10 reference genome for the respective transcriptomic approaches can be found in the following hyperlinks:
mm10 reference genome for spatial transcriptomic
mm10 reference genome for BCBio bulk RNA-seq pipeline
mm10 reference genome for snATAC-seq
mm10 reference genome for split-pipe pipeline. Source data are provided with this paper.

## Code availability
The code used for differential gene expression analysis related to RNA-seq data as well as codes for single nuclei RNA seq using Split-poll barcoding and the single nuclei ATAC seq can be accessed through GitHub[101] (https://github.com/YannVRB/Visium-SOR-mouse).

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

## Acknowledgements

Spatial gene expression using the *10X Genomics Visium* platform was performed at the Iowa NeuroBank Core in the Iowa Neuroscience Institute, and the Genomics Division in the Iowa Institute of Human Genetics which is supported, in part, by the University of Iowa Carver College of Medicine. We thank the Neural Circuits and Behavior Core at the Iowa Neuroscience Institute for use of their facilities. We thank Emily N. Walsh, Jordan Mumm and Savannah Bliese for technical assistance, Dr. Mahesh S. Shetty and Dr. Lisa Lyons for advice on the manuscript. We thank Xiaowen Wang (Partek Inc.) for technical support which was crucial for Visium spatial gene expression data analysis. This work was supported by grants from the National Institute of Health R01 MH 087463 to T.A., The National Institute of Health K99/R00 AG068306 to S.C., and The University of Iowa Hawkeye Intellectual and Developmental Disabilities Research Center (HAWK-IDDRC) P50 HD103556 to T.A. T.A. is also supported by the Roy J. Carver Charitable Trust. J.J.M. is supported by the Roy J. Carver associate professorship.

## Author contributions

S.C. and T.A. conceived the idea. S.C., L.L. and U.M. wrote the manuscript with inputs from all the authors. S.C and U.M. performed viral infusion, S.C., U.M., L.L. and S.E.B. performed behavior, biochemical and molecular biology experiments, analyzed and interpreted the data. Y.V. performed all the bioinformatic analysis, generated plots and analyzed the data with inputs from E.B. and J.J.M. L.C.L. processed the tissue for spatial gene expression.

## Competing interests

T.A. serves on the Scientific Advisory Board of EmbarkNeuro and is a scientific advisor to Aditum Bio and Radius Health. The other authors declare no conflicting interests.
