## [Peer Review File · Nature Communications]

Mapping the spatial transcriptomic signature of the hippocampus during memory consolidationREVIEWER COMMENTS

Reviewer #1 (Remarks to the Author):

Summary: This article uses spatial transcriptomics to profile transcriptional changes in different hippocampal subfields following spatial learning (in a spatial object recognition task). First, the authors collapsed all of their hippocampal transcriptomic data to compare their results to a traditional bulk hippocampal RNA-seq. They show some overlap between gene expression patterns, but some unique transcripts identified via their spatial transcriptomics. They next identify genes up- and down-regulated within each hippocampal subfield and find that the dentate gyrus shows the largest number of genes differentially regulated following learning. Within CA1, a number of genes linked to memory and engram formation were exclusively upregulated, like Nr4a1, Nr4a2, and Nr4a3. Finally, the authors use a viral approach to block Nr4a activity just within CA1 during spatial learning and show that this impairs memory formation. Together, this work presents a nice application of a cutting-edge technique and an important proof-of-concept demonstration that this method works in the brain following behavior, something that is likely to be of broad appeal to the readership of Nature Communications. I do have a few key suggestions for improvement, however, as detailed below.

Comments:

1. It's not clear to me why they chose to include a separate spatial transcriptomics dataset (GEO GSE201601) in Figure 1. Presumably these animals were trained in separate cohorts and might therefore produce batch effects. Was there not enough power using the current experiment alone? It would help to have some evidence that the batches produce similar profiles, both to justify collapsing the data and to demonstrate that spatial transcriptomic profiles are similar across replications. Finally, I can't tell whether they used the combined datasets for the rest of the analyses or whether figure 3 only looks at the current experiment. Providing the n for each experiment (including the bulk RNA-seq experiment) and stating whether both replications were analyzed in Fig. 3 would clarify this.
2. Figure 4 verifies that the Nr4a subfamily is important in CA1 for spatial memory, something that is not surprising, as similar effects have been described before (as detailed in their literature review). They say this functionally validates the spatial transcriptomic analysis, which is somewhat true, but to really know whether the spatial transcriptomics are accurate, it would be more powerful to show that this same manipulation restricted to a region where the Nr4a genes are not induced (CA3 or DG) has no effect. As it stands, it's entirely possible that manipulating Nr4a1/2/3 could potentially impact memory in any brain region. It would be much more convincing if they could show that Nr4a1/2/3 is only important in CA1, not in a neighboring region that doesn't require this gene as indicated by their spatial transcriptomics in figure 3.
3. Is it possible that they see region-specific transcriptional differences simply because CA1, CA3, DG, etc all function on slightly different timelines of gene expression based on their unique afferent and efferent connections?

4. The initial comparison between traditional bulk RNA-seq and spatial transcriptomic “pseudobulk” RNA-seq (Fig. 2) was meant to validate the newer spatial transcriptomic approach but actually showed big differences in the transcriptional profile of the hippocampus after learning (e.g. only 29 pseudobulk genes also came up in the bulk analysis). They discuss how the pseudobulk analysis identified additional genes not identified by the bulk RNA-seq, but was the opposite also true, that the RNA-seq identified genes not considered “hits” in the pseudobulk analysis? Why is the overlap between these techniques so low? Is it due to differences in library construction, sensitivity, analysis, or something else? Some discussion of this discrepancy would be helpful.

Reviewer #2 (Remarks to the Author):

The study conducted by Vanrobeys et al. aimed to uncover the transcriptomic signatures of different subregions of the hippocampus during an early window of spatial memory consolidation. The authors claim to provide the first simultaneous analysis of transcriptomic changes spatially distributed across hippocampal subregions in response to learning using Visium spatial transcriptomics, a high-resolution transcriptomic characterization approach that combines histology and spatial profiling of RNA expression. The researchers define genome-wide transcriptomic changes in the CA1 pyramidal layer, CA1 stratum radiatum, CA1 stratum oriens, CA2+3 pyramidal layer, and dentate gyrus (DG) granular and molecular layers of the dorsal hippocampus. They found that the CA1 pyramidal layer, stratum radiatum, and stratum oriens, as well as the granule cells within the DG, exhibited differential expression patterns for some of the most extensively studied immediate early genes (IEGs) related to transcriptional regulation after spatial exploration. Interestingly, the team found a greater number of differentially expressed genes in DG compared to CA1 following spatial exploration. The study also highlighted transcriptomic signatures within the two relatively understudied hippocampal compartments, stratum radiatum and stratum oriens, which have been challenging to delineate using conventional single-cell sequencing strategies.

The study highlights the spatial diversity in gene expression between subregions, while also demonstrating the crucial mnemonic function of the Nr4a subfamily of transcription factors in the hippocampus.

In short, the study strives to elucidate the transcriptomic diversity that prevails between hippocampal subregions during an early window of spatial memory consolidation, and the molecular mechanisms underlying memory consolidation. The findings of the study could be helpful in defining the mnemonic role of specific hippocampal subregions.

Major concerns

1. Lack of novelty.

The findings presented in this study do not seem to add anything new to the existing literature. Previous research has already shown that learning-induced gene expression occurs in various regions of the hippocampus, including CA1, DG, and individual hippocampal neurons. The authors' own description of the results (e.g. lines 133-140) clearly demonstrate the concerning lack of novelty. Some studies have used TRAP2 to label and examine gene expression in engrams, which allows for engram cell-type specificity. Therefore, it is difficult to understand why such specificity would be sacrificed for spatial

resolution unless it is solely to promote the method used. Furthermore, the use of spatial coordinates is unnecessary when region-specific genetic markers can be used to identify and sort transcriptional profiles from cell types. Moreover, the debut of Visium spatial transcriptomics technique has already been featured in an earlier study that this work builds on. Finally, a dominant negative mouse model of Nr4a transcription factors has already been previously generated and implicated in memory consolidation. Overall, in light of these existing findings, it is simply unclear what new knowledge has been gained from this study.

2. Data reliability issues.

While it is commendable that the authors compared their pseudobulk sequencing approach to traditional bulk sequencing, the fact that they generated different data using each technique raises concerns about the reliability of the results. The authors identified almost twice as many differentially expressed genes (DEGs) using bulk RNA-seq compared to their pseudobulk analysis, which raises questions about the sensitivity and accuracy of their method. Only around 10% of the genes identified with their technique were detected using the traditional approach, and no explanation or attempt at reconciliation was provided. Although the authors did claim that their pseudobulk analysis is more sensitive 'revealing new genes that may be undetectable by other methods', they failed to acknowledge the alternative perspective in which traditional analyses detected 198 more DEGs than their pseudobulk technique. Additionally, the absence of robust c-Fos expression in the pseudobulk data, c-Fos being a positive control, is concerning and warrants an explanation. It is crucial to reconcile the results from different methods and provide a clear rationale for any discrepancies to ensure the reliability of the findings. These discrepancies should also be presented in some form in Figure 2.

3. Relatively scant evidence.

While the study presents an interesting approach to spatial transcriptomics, the evidence provided to support their claims of memory involvement is scant and the data presented is only marginally sufficient. The authors' conclusions appear to be based on minimal evidence, and further experimentation and analysis are necessary to solidify their findings.

4. Inappropriate conclusions drawn from the genetic manipulation experiment.

It is unclear how exactly the manipulation experiment functionally validates their transcriptome dataset. This attempt to draw a direct line of functional validation between their spatial transcriptomic findings and the genetic manipulation experiment appears to be an overreach. The genetic manipulation focused on a single gene, Nr4a, which is already known to have a role in learning and memory, and the manipulation was targeted to a specific region of the hippocampus, the CA1. Despite this highly specific manipulation, the observed effects on memory performance were modest at best. Furthermore, it is challenging to see how this experiment functionally validated the authors' spatial transcriptomics approach, which involves studying the differential gene expression profiles of the entire hippocampus and the participation of over 100 genes. While the experiment is certainly a valuable addition to the literature, it would be more appropriate to view it as complementary rather than conclusive evidence of the authors' transcriptomic findings.

minor notes:

How was the preference score calculated (this is omitted from the methods)? The AAV-Nr4ADN mice actually seem to have a preference for the displaced object and would probably pass a within-group t-test if compared between train vs test performance.

A heat map of the animals activity in the open arena would be a beneficial addition to the data.

1 uL is an excessively large volume used for AAV-infusion (sufficient to fill the entire longitudinal axis of the hippocampus), what is the reason for using such a quantity? Why does expression at the infection site appear inconsistent with the volume used?

5. Communication and data interpretation.

While the study conducted by the authors is interesting, there are several points that need to be addressed in order to support their conclusions more strongly.

a. Firstly, the manuscript title claims to "map the spatial transcriptomic signature of the hippocampus during memory consolidation," yet there is absolutely no evidence suggesting that the animals analyzed actually encoded any memory.

b. The lack of detail provided about the behavioral task used in this study is concerning. The authors describe the task as a spatial object recognition task, but there is no evidence of recognition being demanded from the animals. Instead, the mice were exposed to a novel context for a period of spatial exploration. It is unclear how this type of exposure relates to spatial object recognition or memory consolidation. Moreover, no performance data is provided to support the authors' claims about the behavioral task or the cognitive abilities of the animals. Without more detail about the task and performance data, it is difficult to assess the relevance of the transcriptional signatures identified in this study to memory consolidation or any specific cognitive function. The authors must provide a more detailed description of the task and include performance data in order to substantiate their claims.

c. The authors' descriptions of genetic functions could benefit from greater precision and specificity. Using vague terms such as 'linked to memory' or 'involved in learning' may give an impression of cherry-picking references to fit a predetermined narrative. It may be more appropriate to present a more detailed analysis of the specific genetic functions, and to provide supporting evidence for their role in memory consolidation. Additionally, while some discussion of the functional implications of the results is appropriate, the presentation of such information in the results section may benefit from a more focused and concise description of the key findings. Moreover, some genetic descriptions appear to go against the narrative. E.g. *Usp2* was downregulated in both CA1 and DG subregions. The authors claim *Usp2* is downregulated in the hippocampus following sleep deprivation (lines 211-212), but this implies this DGE is unrelated to memory since one would expect memory to be impaired following sleep deprivation.

d. While the authors suggest that their spatial transcriptomic approach can shed light on the subcellular

localization of RNA as a means of transcriptomic regulation, their claim lacks supporting evidence. For instance, the differential gene expression patterns observed in the stratum oriens and radiatum cannot solely be attributed to dendritic transport of mRNA from CA1 pyramidal neurons. The authors themselves mention that RNA could originate from interneurons and different glial cell types. Such speculations could be limited to the discussion section and omitted from the results. This would help to ensure that the claims made are based on rigorous and reliable scientific evidence.

e. The introduction provides a great comprehensive overview of the research topic, including relevant background on immediate early genes (IEGs), engrams, the canonical trisynaptic pathway, and the subregion-specific function in memory. While the introduction offers valuable insights, the writing style could be improved by avoiding repetition of sentences, which may distract the reader from the scientific content. Additionally, some phrases may appear vague and overly complicated, which can detract from the clarity of the research findings. Therefore, it would be beneficial for the authors to focus on concise writing and clarity of language, enabling the reader to better comprehend the scientific information presented, rather than being distracted by unnecessary repetitions and convoluted phrasing.

An example of unnecessary repetition:

Line 57-58 'Understanding transcriptional dynamics within the hippocampal circuit would provide important insights into ... memory consolidation'

Line 74-76 'Despite the importance of circuitry in the dorsal hippocampus, spatial transcriptomic changes in response to learning across subregions of the dorsal hippocampus remain largely unknown'

Line 77-79 'Learning-induced gene expression ... has not been examined across all subregions simultaneously'.

Line 90-93 'It is still unclear how gene expression in each of the spatially and functionally distinct subregions is regulated after learning. The transcriptomic diversity within these subregions needs to be examined more clearly to better understand the role each of these subregions in memory consolidation'.

f. In Figure 3b hippocampal subregion clusters were presented. Is this data derived from homecage animals, SOR-engaged mice, or does it represent results from the differential expression analysis?

g. What thresholds are being used throughout the study to determine whether a gene is upregulated/downregulated? Are there better ways to convey the extent of expression (instead of describing absolute number of genes) for the data in Figure 3?

Recommendations for improvement

1. Memory consolidation is a complex and dynamic process that involves multiple stages of gene expression. One of the key ways in which the study could be significantly improved and made more interesting for readers of the neuroscientific literature is by tracking additional time points during memory consolidation. While the authors focus on transcriptional events that occur during the (heavily-studied) early wave of genetic expression, which is around 1 hour post-learning, the study could certainly benefit from examining later time points. This would help to provide a more detailed picture of

the complex molecular processes that occur during memory consolidation and could potentially reveal novel insights into the mechanisms underlying memory formation. In this way, the authors will provide a more complete analysis of transcriptional events underlying memory.

2. Secondly, the development of an unbiased transcriptional map of the mouse hippocampus, using genome-scale *in situ* hybridization, has provided detailed molecular evidence for a discretized dorsal–ventral pattern of gene expression. This pattern suggests that genetic domains are not defined by the expression of any single gene but, rather, by the combined overlap of many gene expression domains. The authors could compare the results they obtained in the dorsal hippocampus with the ventral subregions, which may provide some important and novel insights. Multiple segregated molecular subdomains, each containing a unique complement of expressed genes, have been demonstrated along the long axis of the hippocampus (reviewed in Strange et al, 2014). Therefore, exploring the differences in gene expression patterns between the dorsal and ventral hippocampus could provide a more comprehensive understanding of the transcriptional events that occur during memory consolidation.

3. Additionally, the authors could consider incorporating other techniques such as proteomics or epigenomics to gain a more comprehensive understanding of the molecular mechanisms underlying memory consolidation. Such an expansion of the study would significantly enhance the novelty and impact of the findings.

Conclusion

We appreciate the opportunity to review the manuscript. The topic of the study is certainly important and the approach that is taken to examine learning-responsive gene expression across subregions of the hippocampus is promising. However, we regret that the manuscript appears to be premature for publication in *Nature Communications* at this stage. While the study presented in this manuscript is a step in the right direction, it falls short in terms of providing novel insights into the field of memory consolidation. Therefore, it is most crucial that the authors expand the scope of their investigation. In its current state, the manuscript tends to read more like an advertisement for the "Visium Spatial Transcriptomics" technique rather than a rigorous study that provides significant informational value to the scientific community.

While we appreciate the use of unbiased spatial sequencing to elucidate transcriptome-wide changes in gene expression in the hippocampus, the analysis presented is not yet convincing. In particular, the conclusions drawn from the subregion-specific gene expression patterns are not supported by sufficient data. The functional relevance of these patterns remains to be fully demonstrated, and it is unclear how they relate to the observed long-term memory deficits following genetic manipulation of a transcription factor selectively in the CA1 hippocampal subregion.

The manuscript could benefit from more detailed descriptions of the methodology and results. The current presentation is somewhat sparse and may leave readers with questions about how the analyses were conducted and how the data were interpreted.

Based on the above evaluation, it is our opinion that this manuscript is not ready for publication and should therefore be rejected.

Reviewer #3 (Remarks to the Author):

Vanrobeys and colleagues present a compelling manuscript detailing spatial transcriptomic analysis of hippocampal subregions during memory consolidation in mice. The methods presented are novel and their description of analyses will be helpful to other researchers using the 10X Visium platform. The authors first correlated their spatial transcriptomic analysis with traditional bulk RNA-seq to validate the method. Next, they detailed the hippocampal subregion-specific transcriptional changes following learning. Finally, they assessed the functional significance of a candidate DEG using viral manipulation of its expression in vivo. The graphical abstract and figures are clear and concise. This manuscript will be a valuable contribution to the field. My suggestions for revisions are listed below.

Major:

The authors will need to better address sample sizes needed to get a full representation of DEGs. In particular, they should discuss the findings of Schurch (RNA, 2016) and Liu (Bioinformatics, 2014) addressing replicates. Although each spot in the Visium slides might be considered an n, as the authors mention, this topic should be discussed.

Minor:

- Include species and sex of animals in the abstract.
- It would be helpful to show heatmaps of a few key DEGs superimposed on the histology images using Seurat, especially the genes specific to stratum radiatum and/or oriens (eg. Sgk1). Visual representations of the gene expression changes would emphasize the novelty and utility of this technique.
- For comparing bulk-RNA-seq of whole dorsal hippocampal tissue versus “pseudobulk” analysis from Visium, perhaps there would be a greater overlap of genes if the entire region of the hippocampus had been analyzed (including SR & SO of CA2/3 and hilus of the DG)? This would better mimic the amount of tissue analyzed with a bulk dissection of hippocampus.
- Please define “pseudobulk” analysis in greater detail.
- Line 394: For others using 10X Visium platform, it would be helpful to report the mean number of reads per spot and genes per spot, as sequencing depth varies depending on the number of spots covered by tissue within each frame.
- You report that there were 8 brains frozen for spatial transcriptomics – does this mean you used 2 slides with 4 frames each? Please clarify.
- Please consider commenting on the differences in sequencing technologies and how they are analyzed when discussing the results of bulk RNA-seq versus pseudobulk spatial transcriptomics.
- Check verb tenses for consistency across manuscript, for example, line 133 “comprised” instead of present tense “comprise”.
- Line 190 - this sentence does not refer to Figure 4c-d.
- Lines 211-212 : please define “Abeta” as amyloid beta
- Line 323: provide examples of the non-neuronal genes affected by learning
- Line 437: ‘pooled libraries’? or ‘pooled’?

- Line 454: typo, “GO Tern Fusion” should be Term
- Consider showing in situ hybridization for the upregulated Nr4A, without and after learning. It would really make the authors’ case that it is a key upregulated gene, and would give information on what neurons are expressing it. Also, are the neurons expressing any of the Nr4as generally the same neurons expressing Arc or Fos, for example?
- Clarify HCC: were the animals handled like in the SOR? Were they taken out and put back in the HC?

We would like to thank the editor and the reviewers for their positive feedback and insightful comments on how to strengthen the findings of this manuscript.

Based on the reviewers' comments, we have added the following to the revised manuscript:

1. Dentate gyrus-specific manipulation of *Nr4a* transcription function using an AAV-based approach and assessed long-term memory (**Supplementary Figure 8**), thus providing functional relevance of the selective role of *Nr4a* family members in CA1 as revealed by the spatial transcriptomics experiment.
2. A single-nuclei transcriptomics (using a Split-pool barcoding approach) experiment that showed upregulation of *Sgk1* in oligodendrocytes after learning (**Figure 5g-i**).
3. A single-nuclei epigenomics (snATAC-seq) data after learning to show increased promoter accessibility of *Sgk1* in oligodendrocytes (**Figure 5j-l**).
4. RNAscope validation of stratum radiatum- and oriens-specific upregulation of *Sgk1* (**Figure 5c-f**). Heat maps to visualize the expression of *Sgk1* in stratum radiatum and oriens from the spatial transcriptomics data (**Figure 5b**).
5. RNAscope validation of *Nr4a1* and *Nr4a2* expression in CA1 and DG 1 hr after learning (**Supplementary Figure 5a-f**), and *Nr4a1* expression in CA1 and DG 30 mins after learning (**Supplementary Figure 6**). We quantified the colocalization pattern of both activity-induced genes in *Arc*-expressing active neurons (**Supplementary Figure 5g-j**).
6. Low-throughput *Nr4a1* and *Sgk1* gene expression data at 30 min, 1 hr, 2 hr, 6 hr, and 8 hr timepoints after learning (**Figure 2d**), showing that both *Nr4a1* and *Sgk1* are induced within 2 hr after learning.
7. Quadrant plot comparing the transcriptomic profiles between the two batches of spatial gene expression experiments (**Supplementary Figure 2**).
8. Heatmap showing the expression of the top 15 genes from bulk RNA-seq in the pseudobulk samples (**Figure 2c**).
9. Additionally requested behavioral data: heatmaps of habituation (**Supplementary Figure 7a, and Supplementary Figure 8c**), time spent in the inner vs outer zone during habituation (**Supplementary Figure 7b, and Supplementary Figure 8d**), and exploration of objects during the training sessions (**Supplementary fig. 1, Supplementary Figure 7c, and Supplementary Figure 8e**). This shows a correlation between exploration and learned behavior of the spatial location of objects across 3 training sessions.
10. Heatmap showing the expression levels of genes that are upregulated in two or more subregions after learning (**Figure 3g**).
11. A detailed description of the behavioral methodology and results of the new experiments.

A point-by-point response to all reviewers' comments

Note: reviewer's comments are in *italics* and answers are in blue.

Reviewer #1 (Remarks to the Author):

Summary: This article uses spatial transcriptomics to profile transcriptional changes in different hippocampal subfields following spatial learning (in a spatial object recognition task). First, the authors collapsed all of their hippocampal transcriptomic data to compare their results to a traditional bulk hippocampal RNA-seq. They show some overlap between gene expression patterns, but some unique transcripts identified via their spatial transcriptomics. They next identify genes up- and down-regulated within each hippocampal subfield and find that the

dentate gyrus shows the largest number of genes differentially regulated following learning. Within CA1, a number of genes linked to memory and engram formation were exclusively upregulated, like *Nr4a1*, *Nr4a2*, and *Nr4a3*. Finally, the authors use a viral approach to block *Nr4a* activity just within CA1 during spatial learning and show that this impairs memory formation. Together, this work presents a nice application of a cutting-edge technique and an important proof-of-concept demonstration that this method works in the brain following behavior, something that is likely to be of broad appeal to the readership of Nature Communications. I do have a few key suggestions for improvement, however, as detailed below.

We thank reviewer #1 for their thorough evaluation and positive comments.

1. It's not clear to me why they chose to include a separate spatial transcriptomics dataset (GEO GSE201601) in Figure 1. Presumably these animals were trained in separate cohorts and might therefore produce batch effects. Was there not enough power using the current experiment alone? It would help to have some evidence that the batches produce similar profiles, both to justify collapsing the data and to demonstrate that spatial transcriptomic profiles are similar across replications.

We chose to include a separate spatial transcriptomics dataset from our recent work to enhance the rigor of our data. Both were performed using the same spatial learning task and sampled at the 1 hr time point. This increased our biological replications from n=4 to n=7 per group, improving our statistical power to detect differentially expressed genes (DEGs).

Based on this comment from Reviewer #1, we performed a correlation analysis of the transcriptomic profiles to identify any differences between the two batches. The results of this analysis are shown in **Supplementary Figure S2** in the revised manuscript. The data show a highly significant correlation ($p < 0.00001$) between the differentially expressed genes (DEGs) from the two batches, suggesting the two cohorts of mice exhibit similar changes in transcriptomic profiles.

Finally, I can't tell whether they used the combined datasets for the rest of the analyses or whether figure 3 only looks at the current experiment. Providing the n for each experiment (including the bulk RNA-seq experiment) and stating whether both replications were analyzed in Fig. 3 would clarify this.

Both datasets (n=7) were combined in the data provided in Figures 1-3. We have made this clear in the revised manuscript. We used n=4 per group in our bulk RNA seq experiment.

2. Figure 4 verifies that the *Nr4a* subfamily is important in CA1 for spatial memory, something that is not surprising, as similar effects have been described before (as detailed in their literature review). They say this functionally validates the spatial transcriptomic analysis, which is somewhat true, but to really know whether the spatial transcriptomics are accurate, it would be more powerful to show that this same manipulation restricted to a region where the *Nr4a* genes are not induced (CA3 or DG) has no effect.

As it stands, it's entirely possible that manipulating *Nr4a* 1/2/3 could potentially impact memory in any brain region. It would be much more convincing if they could show that *Nr4a* 1/2/3 is only important in CA1, not in a neighboring region that doesn't require this gene as indicated by their spatial transcriptomics in figure 3.

To validate our spatial transcriptomics analysis, we performed RNAscope *in situ* hybridization to detect the expression of *Nr4a* family members in the dorsal hippocampus. We show that both *Nr4a1* and *Nr4a2* genes are upregulated in CA1 but not in DG following learning (**Supplementary Figure 5**). We then manipulated *Nr4a* function in the DG, a region where the *Nr4a* genes are not induced following learning (**Supplementary Figure 8**). We found that DG-

specific manipulation of Nr4a function did not affect spatial memory consolidation (**Supplementary Figure 8d**). This supports our spatial transcriptomic analysis and underscores the important role of the Nr4a function in CA1.

3. Is it possible that they see region-specific transcriptional differences simply because CA1, CA3, DG, etc all function on slightly different timelines of gene expression based on their unique afferent and efferent connections?

Immediate-early gene expression has been reported to be induced with a similar time course in different hippocampal regions. For example, *Arc* is induced in both CA1 and CA3 only at early time points after learning (Vazdarjanova et al., 2002). Thus, it appears that *Arc* is temporally and briefly upregulated by neuronal activity with a similar time course in different hippocampal subfields (Pevzner et al., 2012). To explore this issue, we have added additional data to the manuscript. Using RNAscope, we found that *Arc* was induced with *Nr4a1* and *Nr4a2* in CA1 1 hr after learning (**Supplementary Figure 5g-j**). Similarly, *Nr4a1* induction was seen in CA1 at 30 min and 1 hr after learning but not in DG at these time points (**Supplementary Figure 5 and 6**). We also show new low-throughput gene expression data from the dorsal hippocampus following different time points after learning (0.5, 1, 2, 6, and 8 hrs), showing that learning-induced genes are upregulated only in a short early temporal window (**Figure 2d**). Thus, it does not appear that there are different timelines of gene expression in different hippocampal subfields, but future experiments to examine other specific genes will be necessary to explore this issue more completely.

4. The initial comparison between traditional bulk RNA-seq and spatial transcriptomic “pseudobulk” RNA-seq (Fig. 2) was meant to validate the newer spatial transcriptomic approach but actually showed big differences in the transcriptional profile of the hippocampus after learning (e.g., only 29 pseudobulk genes also came up in the bulk analysis). They discuss how the pseudobulk analysis identified additional genes not identified by the bulk RNA-seq, but was the opposite also true, that the RNA-seq identified genes not considered “hits” in the pseudobulk analysis? Why is the overlap between these techniques so low? Is it due to differences in library construction, sensitivity, analysis, or something else? Some discussion of this discrepancy would be helpful.

Bulk RNA-seq seemed like the most accurate unbiased sequencing approach to compare to ‘pseudobulk’ RNA-seq of spatial transcriptomics. We did not expect the transcriptomic profiles of these two techniques to be identical, as no two sequencing techniques have the same approach to detect DEGs. Bulk RNA-seq included all transcripts, except ribosomal RNA, within all subregions and cell types in the whole dorsal hippocampus. In contrast, pseudobulk spatial RNA-seq only included poly-A mRNAs within selected barcoded spots on the principal cell layers, stratum radiatum and oriens in a 10 µm brain slice. Additional differences in overlap can be attributed to the type of statistical test used to compute ‘significant’ changes in gene expression. Pseudobulk was analyzed by rank-sum Kruskal-Wallis test with a 1.4 threshold for fold-change. Bulk RNA-seq was analyzed by the conventional EdgeR statistical test. This approach would be inappropriate for our pseudobulk analysis because it utilizes the mean instead of the median to compute fold-change and would increase the number of false positive “hits.” Therefore, some DEGs in bulk RNA-seq appeared as a low fold-change in pseudobulk because they did not surpass the 1.4 threshold for the Kruskal-Wallis test and are not considered “hits”. DEGs identified by bulk RNA-seq and not pseudobulk are related to transcription regulation (*Fos*, *Egr2*, *Hif3a*, *Fosl2*, *Nfkb1a*), protein processing in ER (*Hspa1b*, *Hspa1a*, *Herpud1*, *Pdia6*, *Pdia4*, *Hsph1*) and protein kinase regulator activity (*Hspb1*, *Cdkn1a*, *Trib1*, *Cables1*). Even still, both techniques can be used to understand the transcriptomic

changes in the dorsal hippocampus after learning – pseudobulk across subregions in a given coronal section and bulk RNA-seq in all tissue and cells. These discrepancies in overlap are discussed in the revised manuscript and illustrated in a heatmap of the pseudobulk analysis for the top 15 DEGs in bulk RNA-seq (Figure 2c).

Reviewer #2 (Remarks to the Author):

The study conducted by Vanrobeys et al. aimed to uncover the transcriptomic signatures of different subregions of the hippocampus during an early window of spatial memory consolidation. The authors claim to provide the first simultaneous analysis of transcriptomic changes spatially distributed across hippocampal subregions in response to learning using Visium spatial transcriptomics, a high-resolution transcriptomic characterization approach that combines histology and spatial profiling of RNA expression. The researchers define genome-wide transcriptomic changes in the CA1 pyramidal layer, CA1 stratum radiatum, CA1 stratum oriens, CA2+3 pyramidal layer, and dentate gyrus (DG) granular and molecular layers of the dorsal hippocampus. They found that the CA1 pyramidal layer, stratum radiatum, and stratum oriens, as well as the granule cells within the DG, exhibited differential expression patterns for some of the most extensively studied immediate early genes (IEGs) related to transcriptional regulation after spatial exploration. Interestingly, the team found a greater number of differentially expressed genes in DG compared to CA1 following spatial exploration. The study also highlighted transcriptomic signatures within the two relatively understudied hippocampal compartments, stratum radiatum and stratum oriens, which have been challenging to delineate using conventional single-cell sequencing strategies.

The study highlights the spatial diversity in gene expression between subregions, while also demonstrating the crucial mnemonic function of the Nr4a subfamily of transcription factors in the hippocampus.

In short, the study strives to elucidate the transcriptomic diversity that prevails between hippocampal subregions during an early window of spatial memory consolidation, and the molecular mechanisms underlying memory consolidation. The findings of the study could be helpful in defining the mnemonic role of specific hippocampal subregions.

We thank reviewer #2 for their insightful suggestions for improvement. We have addressed their comments and provided the rationale for revisions to our manuscript below.

1. Lack of novelty.

The findings presented in this study do not seem to add anything new to the existing literature. Previous research has already shown that learning-induced gene expression occurs in various regions of the hippocampus, including CA1, DG, and individual hippocampal neurons. The authors' own description of the results (e.g. lines 133-140) clearly demonstrate the concerning lack of novelty. Some studies have used TRAP2 to label and examine gene expression in engrams, which allows for engram cell-type specificity. Therefore, it is difficult to understand why such specificity would be sacrificed for spatial resolution unless it is solely to promote the method used. Furthermore, the use of spatial coordinates is unnecessary when region-specific genetic markers can be used to identify and sort transcriptional profiles from cell types.

We appreciate the opportunity to clarify this important point and elaborate on the areas of our study that contribute novel information to the literature.

1) Differential gene expression in CA1 stratum radiatum and oriens after learning. We are the first to elucidate the gene expression patterns represented within the CA1 stratum radiatum and oriens after learning. We provide new evidence to support the relationship between these

components of area CA1 in the early stages of memory consolidation. This finding is significant because stratum radiatum and oriens cannot be studied by single-cell/nuclei or TRAP-seq approach due to a lack of marker genes. In the revised manuscript, we identified oligodendrocyte-specific upregulation of *Sgk1* in stratum radiatum and oriens surrounding the CA1 pyramidal layer (**Figure 5**). This combined approach demonstrates the power of spatial transcriptomics to elucidate novel gene expression signatures.

2) Transcriptomic signature of engram-specific genes across hippocampal subregions.

Spatial transcriptomics is a new approach to studying gene expression from a brain slice without compromising the tissue anatomy. Spatial transcriptomics is a very comprehensive, unbiased sequencing technique, which is why we chose this approach to study the spatial regulation of engram-specific genes after learning. Instead of using engram-specific markers, we could define engrams by the loci that express DEGs in response to learning within the hippocampus without mechanical dissociation or removing the afferent and efferent connections between subregions. This enabled us to define unbiased expression of engram-specific genes at a high spatial resolution.

3) CA1 subregion-specific function of the Nr4a family of transcription factors during memory consolidation.

Our study shows that blocking the function of the Nr4a transcription factors exclusively in CA1 impairs memory, while similar manipulation in DG has no effect on memory (**Supplementary Figure 8**). This subregion-specific manipulation and its direct effects on memory consolidation have not been performed previously. Nr4a regulates a number of downstream genes and contributes to synaptic function in the pyramidal neurons of the hippocampus. Our findings demonstrate that its role in memory is defined by its spatial expression within CA1 of the hippocampus.

Moreover, the debut of Visium spatial transcriptomics technique has already been featured in an earlier study that this work builds on.

Our previous study (Bahl E, 2022) used a computational tool (NEUROeSTIMator) to predict patterns of neuronal activity across the whole brain. Bahl et al. did not investigate DEGs within hippocampal subregions, nor did they use spatial transcriptomics to generate information regarding which genes were up- and downregulated following learning. In the current study, we applied spatial transcriptomics to define the gene expression signature within hippocampal subregions after learning.

Finally, a dominant negative mouse model of Nr4a transcription factors has already been previously generated and implicated in memory consolidation. Overall, in light of these existing findings, it is simply unclear what new knowledge has been gained from this study.

In this study, we infused a dominant negative construct of Nr4a (AAV-Nr4ADN) exclusively within the hippocampal CA1 subregion. The previously described Nr4ADN mouse model (Chatterjee et al., 2022; Hawk et al., 2012) uses the tTA and tetO system to drive the expression of Nr4ADN in all excitatory neurons of the forebrain. The Nr4aDN mouse model focused on the role of Nr4a within the forebrain during memory consolidation, not within a specific hippocampal subregion.

In our revised manuscript, we extend our work to DG-specific manipulation of Nr4a transcription function and assessed long-term memory (**Supplementary Figure 8**). We found that the function of Nr4a transcription factors in the DG does not contribute to spatial memory consolidation. These new results demonstrate the subregion specific contribution of these immediate-early gene products to memory consolidation.

2. Data reliability issues.

While it is commendable that the authors compared their pseudobulk sequencing approach to

traditional bulk sequencing, the fact that they generated different data using each technique raises concerns about the reliability of the results. The authors identified almost twice as many differentially expressed genes (DEGs) using bulk RNA-seq compared to their pseudobulk analysis, which raises questions about the sensitivity and accuracy of their method. Only around 10% of the genes identified with their technique were detected using the traditional approach, and no explanation or attempt at reconciliation was provided. Although the authors did claim that their pseudobulk analysis is more sensitive ‘revealing new genes that may be undetectable by other methods’, they failed to acknowledge the alternative perspective in which traditional analyses detected 198 more DEGs than their pseudobulk technique.

Pseudobulk and bulk RNA-seq are expected to generate slightly different lists of genes, as they take a different approach to study gene expression. We expected some overlap, but no two sequencing techniques will produce identical results, and can still be reliable methods. We identified 224 DEGs from bulk RNA-seq compared to 119 DEGs of pseudobulk RNA-seq. This discrepancy could be because bulk RNA seq included the whole transcriptome, except ribosomal RNA, within all cell types and subregions of the entire dorsal hippocampus. On the contrary, pseudobulk RNA-seq only included poly-A mRNAs within selected barcoded spots on the principal cell layers, stratum radiatum, and oriens in a 10 μ m brain slice. Any additional differences in overlap can be attributed to the type of statistical test used to compute ‘significant’ changes in gene expression. Pseudobulk was analyzed by rank-sum Kruskal-Wallis test. With this algorithm, the median computes fold change to remain consistent across conditions. Bulk RNA-seq was analyzed by the conventional EdgeR statistical test. This approach would be in appropriate for our pseudobulk analysis because it utilizes the mean instead of the median to compute fold-change and would increase the number of false positive DEGs. Some DEGs in bulk RNA-seq appeared as a low fold-change in pseudobulk because they did not surpass the 1.4 threshold for the Kruskal-Wallis test and are not considered DEGs. Despite the use of two different statistical tests, 30 genes overlapped between the two techniques with a significant correlation between the gene expression changes (p value <0.0001). We chose to compare our pseudobulk data with bulk RNA-seq data as this is a standard approach to quantifying RNA enrichment within a sample, but the differences in how the libraries were prepared, the amount of tissue and subregions sequenced, and the statistical approach explain why we observe differences in gene expression. The discrepancy in overlap is discussed in the revised manuscript.

Additionally, the absence of robust c-Fos expression in the pseudobulk data, c-Fos being a positive control, is concerning and warrants an explanation. It is crucial to reconcile the results from different methods and provide a clear rationale for any discrepancies to ensure the reliability of the findings. These discrepancies should also be presented in some form in Figure 2.

c-Fos (FDR 5.37 E-13) was upregulated in pseudobulk after learning, but above the 1.4 threshold used to compute fold-change for the Kruskal-Wallis test. We have presented this discrepancy in a heatmap of the pseudobulk analysis for the top 15 DEGs in bulk RNA-seq (Figure 2c). Indeed, we found significant induction of c-Fos in CA2 and CA3 subregions after learning (Fig 3f-g), overlapping with several IEGs including Erg1, Arc, and Nr4a2. In terms of a positive control, we observed robust Arc expression in both pseudobulk and bulk RNA-seq (Figure 2b). Arc is a well-defined marker for neuronal activity, and we see expression of Nr4a1

and *Nr4a2* in *Arc*-positive neurons after learning using RNAscope approach in new experiments added to the revised manuscript (**Supplemental Figure 5g-j**).

3. Relatively scant evidence.

While the study presents an interesting approach to spatial transcriptomics, the evidence provided to support their claims of memory involvement is scant and the data presented is only marginally sufficient. The authors' conclusions appear to be based on minimal evidence, and further experimentation and analysis are necessary to solidify their findings.

The reviewer brings up an important point. We were unable to test the memory of mice that were used for the spatial transcriptomics experiment as they were euthanized 1 hr after training. However, the spatial object recognition (SOR) task is a well-established paradigm used to study learning-induced gene expression (Chatterjee et al., 2022; Kwapis et al., 2018; Vogel-Ciernia & Wood, 2014). To provide additional evidence of memory involvement, we provide new analysis of SOR training for mice used in the spatial transcriptomics experiment. The mice display a progressive decrease in exploration towards objects across training trials, suggesting the acquisition of information about the objects (**Supplementary Figure 1a, 6b and 7c**). We also showed effective memory consolidation using the same SOR paradigm in our *Nr4a*-manipulation experiment (**Figure 4**). The control mice (eGFP infused in CA1 or DG) exhibit memory of the original location of the displaced object during the 24-hour retention test (**Figure 4e, Supplementary Figure 8f**). These additional behavioral analyses strongly support the involvement of memory in our paradigm used for spatial transcriptomic experiments. Further, the functional importance of genes identified using this technique—for example the specific upregulation of *Nr4a* genes in CA1 relative to DG—is now clearly demonstrated in the revised manuscript, thus showing the the genes identified are important for memory itself.

4. Inappropriate conclusions drawn from the genetic manipulation experiment.

*It is unclear how exactly the manipulation experiment functionally validates their transcriptome dataset. This attempt to draw a direct line of functional validation between their spatial transcriptomic findings and the genetic manipulation experiment appears to be an overreach. The genetic manipulation focused on a single gene, *Nr4a*, which is already known to have a role in learning and memory, and the manipulation was targeted to a specific region of the hippocampus, the CA1. Despite this highly specific manipulation, the observed effects on memory performance were modest at best. Furthermore, it is challenging to see how this experiment functionally validated the authors' spatial transcriptomics approach, which involves studying the differential gene expression profiles of the entire hippocampus and the participation of over 100 genes. While the experiment is certainly a valuable addition to the literature, it would be more appropriate to view it as complementary rather than conclusive evidence of the authors' transcriptomic findings.*

The manipulation of the *Nr4a* transcription factors in CA1 and DG did not serve to validate the entire spatial transcriptomics dataset. We agree the *Nr4ADN* viral infusion and behavioral experiments complement our spatial transcriptomics findings, and this has been rephrased in the revised manuscript. It would not be possible to functionally validate the entire dataset in this study. Our data instead provides information for future studies to address the functional relevance of their differential expression after learning. Rather, we demonstrated the role of 3 genes (*Nr4a1*, *Nr4a2* and *Nr4a3*) at once by functional manipulation in CA1 and DG and several genes by *in situ* hybridization approach. We showed that the *Nr4a* transcription factors have direct roles in CA1 subregion for memory consolidation (**Figure 4**), in support of their subregion-specific expression in CA1 shown by spatial transcriptomics and RNAscope *in situ* hybridization (**Figure 5, Supplemental Figure 6**).

How was the preference score calculated (this is omitted from the methods)?

$$\text{Percent preference for displaced object} = \frac{(\text{exploration towards the displaced object})}{(\text{total exploration towards all objects})} \times 100$$

The AAV-Nr4ADN mice actually seem to have a preference for the displaced object and would probably pass a within-group t-test if compared between train vs test performance.

A within-group unpaired t-test comparing the training and test performance of Nr4ADN mice showed a slight tendency towards memory performance (p=0.051). However, a t-test is not recommended for SOR data analysis as it cannot compare with the performance of the control mice during the same task. The more appropriate statistical test is a two-way ANOVA. The CA1-specific Nr4ADN mice did not pass the 0.05 p-value threshold for training vs test performance but performed significantly lower compared to control mice during test performance (**p=0.0014). This reduced performance shows that loss of Nr4a function in CA1 impairs memory consolidation.

A heat map of the animal's activity in the open arena would be a beneficial addition to the data.

We have provided open-field activity data in the revised manuscript. We show reference heatmaps during habituation and graphs showing the percent time spent in the outer vs inner zone of the open field during the habituation session (**Supplementary Figure 7a-b** and **Supplementary Figure 8c,d**).

1 uL is an excessively large volume used for AAV-infusion (sufficient to fill the entire longitudinal axis of the hippocampus), what is the reason for using such a quantity? Why does expression at the infection site appear inconsistent with the volume used?

We used a specific AAV serotype (AAV2/2), which does not diffuse across subregions, to confine viral infusion only within the CA1 (**Figure 4c**). Our strategy to express the Nr4ADN construct exclusively within a single subregion was verified using immunofluorescence from brain slices (**Figure 4c**).

5. Communication and data interpretation.

While the study conducted by the authors is interesting, there are several points that need to be addressed in order to support their conclusions more strongly.

a. Firstly, the manuscript title claims to "map the spatial transcriptomic signature of the hippocampus during memory consolidation," yet there is absolutely no evidence suggesting that the animals analyzed encoded any memory.

We cannot show that the mice used for spatial transcriptomics encoded memory because their brains were harvested 1 hr following the training session. However, we show new analysis to support these mice learning the location of the objects as evident from decreased exploration across training trials (**Supplementary figure 1**) and would display memory if tested 24 h later, as shown by the control mice trained in the same behavioral paradigm (eGFP mice in **Fig 4e**, **Supplementary Figure 8f**). Importantly, we found engram-related genes induced in hippocampal subregions, providing evidence that the mice used for spatial transcriptomics were in the early stages of memory consolidation.

b. The lack of detail provided about the behavioral task used in this study is concerning. The authors describe the task as a spatial object recognition task, but there is no evidence of recognition being demanded from the animals. Instead, the mice were exposed to a novel context for a period of spatial exploration. It is unclear how this type of exposure relates to spatial object recognition or memory consolidation. Moreover, no performance data is provided

to support the authors' claims about the behavioral task or the cognitive abilities of the animals. Without more detail about the task and performance data, it is difficult to assess the relevance of the transcriptional signatures identified in this study to memory consolidation or any specific cognitive function. The authors must provide a more detailed description of the task and include performance data in order to substantiate their claims.

We added the requested details about the SOR task to the methods section of the revised manuscript. Although the mice were exposed to novel objects, the objects were placed in specific spatial locations respective to a spatial cue in the arena. Therefore, learning the location of the objects in reference to the spatial cue distinguishes this paradigm from a novelty task to facilitate memory consolidation (Chatterjee et al., 2022; Kwapis et al., 2018; Vogel-Ciernia & Wood, 2014).

We also provide the requested performance data (**Supplemental Figure 1**) in the revised manuscript. Decreased exploration of the objects across 3 training sessions indicates the mice have learned the spatial location of the objects (**Supplemental Figure 1**). Mice that have consolidated the memory of the spatial locations of these objects spend more time exploring the displaced object 24 hr later during the test session (control GFP mice: **Figure 4e and Supplementary Figure 8d**). This behavioral analysis confirms that the SOR paradigm relates to the consolidation of long-term memory and is relevant to use to study learning-induced gene expression in this study.

*c. The authors' descriptions of genetic functions could benefit from greater precision and specificity. Using vague terms such as 'linked to memory' or 'involved in learning' may give an impression of cherry-picking references to fit a predetermined narrative. It may be more appropriate to present a more detailed analysis of the specific genetic functions, and to provide supporting evidence for their role in memory consolidation. Additionally, while some discussion of the functional implications of the results is appropriate, the presentation of such information in the results section may benefit from a more focused and concise description of the key findings. Moreover, some genetic descriptions appear to go against the narrative. E.g. *Usp2* was downregulated in both CA1 and DG subregions. The authors claim *Usp2* is downregulated in the hippocampus following sleep deprivation (lines 211-212), but this implies this DGE is unrelated to memory since one would expect memory to be impaired following sleep deprivation.*

We refocused our descriptions of the DEGs detected by spatial transcriptomics in the revised manuscript as requested. We have cited and included relevant information from the literature about their involvement in the hippocampus or role in memory consolidation. The DEGs described with terms 'linked to memory' or 'involved in learning' is appropriate because their role in memory is less understood. Due to their differential expression compared to HC mice, we can suggest they have a role in the cellular functions that occur in the hippocampus following learning. As suggested, we have removed the description of the role of *Usp2* in sleep deprivation. As the current study does not involve sleep, the reference cited is not appropriate to the current study.

d. While the authors suggest that their spatial transcriptomic approach can shed light on the subcellular localization of RNA as a means of transcriptomic regulation, their claim lacks supporting evidence. For instance, the differential gene expression patterns observed in the stratum oriens and radiatum cannot solely be attributed to dendritic transport of mRNA from CA1 pyramidal neurons. The authors themselves mention that RNA could originate from interneurons and different glial cell types. Such speculations could be limited to the discussion section and omitted from the results. This would help to ensure that the claims made are based on rigorous and reliable scientific evidence.

We provided this mechanism as a suggested outcome of DEGs detected in stratum oriens and radiatum, because spatial transcriptomics can detect all RNA within cells, including RNA localized to the dendrites of neurons innervating other subregions. We discovered that *Sgk1* is upregulated in oligodendrocytes of stratum radiatum and oriens (**Figure 5**). This finding is now supported by single nuclei multiomics in the revised manuscript.

e. The introduction provides a great comprehensive overview of the research topic, including relevant background on immediate early genes (IEGs), engrams, the canonical trisynaptic pathway, and the subregion-specific function in memory. While the introduction offers valuable insights, the writing style could be improved by avoiding repetition of sentences, which may distract the reader from the scientific content. Additionally, some phrases may appear vague and overly complicated, which can detract from the clarity of the research findings. Therefore, it would be beneficial for the authors to focus on concise writing and clarity of language, enabling the reader to better comprehend the scientific information presented, rather than being distracted by unnecessary repetitions and convoluted phrasing.

An example of unnecessary repetition:

Line 57-58 'Understanding transcriptional dynamics within the hippocampal circuit would provide important insights into ... memory consolidation'

Line 74-76 'Despite the importance of circuitry in the dorsal hippocampus, spatial transcriptomic changes in response to learning across subregions of the dorsal hippocampus remain largely unknown'

Line 77-79 'Learning-induced gene expression ... has not been examined across all subregions simultaneously'.

Line 90-93 'It is still unclear how gene expression in each of the spatially and functionally distinct subregions is regulated after learning. The transcriptomic diversity within these subregions needs to be examined more clearly to better understand the role each of these subregions in memory consolidation'.

We edited repetitive sentences, including those listed above, as well as our wording to be concise and easy to comprehend in the revised manuscript.

f. In Figure 3b hippocampal subregion clusters were presented. Is this data derived from homecage animals, SOR-engaged mice, or does it represent results from the differential expression analysis?

The hippocampal subregion clusters presented in Figure 3b are derived from all animals (7 HC vs 7 SOR).

g. What thresholds are being used throughout the study to determine whether a gene is upregulated/downregulated? Are there better ways to convey the extent of expression (instead of describing absolute number of genes) for the data in Figure 3?

We used a 1.4 threshold for fold-change and 0.05 FDR to determine significant DEGs. The revised manuscript includes a heat map to show the extent of gene expression changes between different hippocampal subregions after learning (**Figure 3g**).

Recommendations for improvement

1. Memory consolidation is a complex and dynamic process that involves multiple stages of gene expression. One of the key ways in which the study could be significantly improved and made more interesting for readers of the neuroscientific literature is by tracking additional time points during memory consolidation. While the authors focus on transcriptional events that occur

during the (heavily-studied) early wave of genetic expression, which is around 1 hour post-learning, the study could certainly benefit from examining later time points. This would help to provide a more detailed picture of the complex molecular processes that occur during memory consolidation and could potentially reveal novel insights into the mechanisms underlying memory formation. In this way, the authors will provide a more complete analysis of transcriptional events underlying memory.

Our study focused on the early induction of gene expression because this is arguably the most important timeframe to understand how learning is transformed into long-term memories (hence why it is heavily studied). This early induction, or first wave, of learning-induced gene expression within each subregion has not been simultaneously studied using an unbiased, comprehensive approach like spatial transcriptomics. Within 1 hr, we see global upregulation of IEGs *Egr1* and *Homer1*, and subregion-specific upregulation of *Nr4a1*, *Dusp5*, *Arc*, and *Sgk1*. In new experiments added to the revised manuscript, we looked at later timepoints (2, 6, and 8 hrs), and found that *Nr4a1* and *Sgk1* are not induced in the hippocampus (**Figure 2d**). This is congruent with the literature that gene expression at timepoints beyond 2 hr is minimal and has a less prominent role in memory consolidation (Peixoto et al 2015, Poplawski et al 2016). Thus, examining the spatial gene expression at later time points is beyond the scope of the current study and is suggested as a future direction in the Discussion.

2. Secondly, the development of an unbiased transcriptional map of the mouse hippocampus, using genome-scale *in situ* hybridization, has provided detailed molecular evidence for a discretized dorsal–ventral pattern of gene expression. This pattern suggests that genetic domains are not defined by the expression of any single gene but, rather, by the combined overlap of many gene expression domains. The authors could compare the results they obtained in the dorsal hippocampus with the ventral subregions, which may provide some important and novel insights. Multiple segregated molecular subdomains, each containing a unique complement of expressed genes, have been demonstrated along the long axis of the hippocampus (reviewed in Strange et al, 2014). Therefore, exploring the differences in gene expression patterns between the dorsal and ventral hippocampus could provide a more comprehensive understanding of the transcriptional events that occur during memory consolidation.

We disagree about the relevance and novel insights that would be obtained from the ventral hippocampus in this study. Manipulation of the ventral hippocampus by lesion or loss-of-function approaches has no effects on long-term spatial memory (Hock & Bunsey, 1998; Moser et al., 1993; Moser et al., 1995).

3. Additionally, the authors could consider incorporating other techniques such as proteomics or epigenomics to gain a more comprehensive understanding of the molecular mechanisms underlying memory consolidation. Such an expansion of the study would significantly enhance the novelty and impact of the findings.

In the revised manuscript, we provide new single-nuclei split-pool barcoding data to identify the cell type responsible for *Sgk1* expression in CA1 stratum radiatum and oriens (**Figure 5g-i**). Further, we have added data using epigenomics. We show single-nuclei ATAC-seq data to elucidate cell type-specific chromatin accessibility of *Sgk1* promoter 1 hr after learning (**Figure 5j-l**). We discovered that *Sgk1* is induced by learning only within oligodendrocytes of stratum radiatum and oriens, which functionally and structurally support the axons of CA1 pyramidal cells.

Conclusion

We appreciate the opportunity to review the manuscript. The topic of the study is certainly important and the approach that is taken to examine learning-responsive gene expression

across subregions of the hippocampus is promising. However, we regret that the manuscript appears to be premature for publication in Nature Communications at this stage. While the study presented in this manuscript is a step in the right direction, it falls short in terms of providing novel insights into the field of memory consolidation. Therefore, it is most crucial that the authors expand the scope of their investigation. In its current state, the manuscript tends to read more like an advertisement for the "Visium Spatial Transcriptomics" technique rather than a rigorous study that provides significant informational value to the scientific community. While we appreciate the use of unbiased spatial sequencing to elucidate transcriptome-wide changes in gene expression in the hippocampus, the analysis presented is not yet convincing. In particular, the conclusions drawn from the subregion-specific gene expression patterns are not supported by sufficient data. The functional relevance of these patterns remains to be fully demonstrated, and it is unclear how they relate to the observed long-term memory deficits following genetic manipulation of a transcription factor selectively in the CA1 hippocampal subregion. The manuscript could benefit from more detailed descriptions of the methodology and results. The current presentation is somewhat sparse and may leave readers with questions about how the analyses were conducted and how the data were interpreted. Based on the above evaluation, it is our opinion that this manuscript is not ready for publication and should therefore be rejected.

Novel insights into the field of memory: We have shown gene expression changes induced by learning simultaneously across critical hippocampal subregions of a tissue slice for the first time. This includes the understudied subregions stratum radiatum and oriens, which are highly interconnected with CA1 and little was known of their transcriptomic signature after learning until now. We discovered *Sgk1* expression in oligodendrocytes of stratum radiatum and oriens, which contributes to our understanding of how these subregions may support CA1 in memory consolidation. Additionally, we demonstrated that a subfamily of transcription factors can have subregion-specific functions within the hippocampus to encode long-term memory. This has expanded our current understanding in the field of memory of how transcriptional changes drive memory consolidation in the hippocampus.

Spatial transcriptomics provides significant information of value to the science community: In addition to spatial transcriptomics being a new technique to study unbiased, large-scale changes in gene expression, we are the first to apply this approach to study differential gene expression after learning in subregions of the hippocampus. Spatial transcriptomics is unique in that it defines DEGs by the anatomical boundaries of intact tissue samples, eliminating the need for marker genes or IEG expression to first define engram ensembles. This unbiased, cutting-edge approach cannot be performed by single-nuclei, single-cell, or TRAP sequencing techniques. Stratum radiatum and oriens cannot be detected by any other sequencing technique because they do not express specific marker genes distinct from CA1, CA2/3 or DG. Therefore, engram cell-type specificity has not been studied in these subregions surrounding CA1, until now. We have shown that spatial transcriptomics can be used to study previously unidentifiable subregions that do not display region-specific markers but are known to contribute to learning and memory consolidation, thus providing valuable information to the scientific community.

We have **expanded the scope of our study** with new experiments and data analysis, adjusted our claims and conclusions to be **supported by the data presented**, provided **detailed descriptions of methodology** in the methods section, and additional information regarding **data analysis and interpretation** for a clear understanding by readers. We believe the additional experiments, information, and our novel approach to studying learning-induced gene expression at a subregion-specific level greatly improves our study.

Reviewer #3 (Remarks to the Author):

Vanrobeys and colleagues present a compelling manuscript detailing spatial transcriptomic analysis of hippocampal subregions during memory consolidation in mice. The methods presented are novel and their description of analyses will be helpful to other researchers using the 10X Visium platform. The authors first correlated their spatial transcriptomic analysis with traditional bulk RNA-seq to validate the method. Next, they detailed the hippocampal subregion-specific transcriptional changes following learning. Finally, they assessed the functional significance of a candidate DEG using viral manipulation of its expression in vivo. The graphical abstract and figures are clear and concise. This manuscript will be a valuable contribution to the field. My suggestions for revisions are listed below.

We thank reviewer #3 for their positive evaluation. We have addressed their comments and provided rationale for revisions to our manuscript below.

1. The authors will need to better address sample sizes needed to get a full representation of DEGs. In particular, they should discuss the findings of Schurch (RNA, 2016) and Liu (Bioinformatics, 2014) addressing replicates.

Our spatial transcriptomics sample size was 14 (7 HC vs 7 SOR). We began with an initial cohort of 8 biological replicates (4 HC vs 4 SOR). Liu et al. found that increasing the number of replicates improved the reproducibility of the results and reduced the false discovery rate (Liu et al., 2014). They also found that a minimum of three replicates per condition was required to achieve reasonable statistical power. Previous spatial transcriptomic studies have used 3 biological replicates (Stahl et al., 2016) and 8 biological replicates (Vickovic et al., 2019). However, Schurch et al. also found that increasing the number of replicates significantly improved the power to detect DEGs by RNA sequencing and reduced the false discovery rate (Schurch et al., 2016). They showed that sample size was more important than sequencing depth to accurately detect DEGs. Because of this, we included 6 additional replicates (3 HC vs 3 SOR) from a recent study (Bahl E, 2022) to improve the statistical power to detect DEGs by spatial transcriptomics. This has been addressed in the results and the methods section of the revised manuscript.

2. Although each spot in the Visium slides might be considered an n, as the authors mention, this topic should be discussed.

The spots assigned to the hippocampus of a Visium slide are spaced apart from each other (i.e., do not overlap) and correspond to a unique set of cells within the tissue sample. Therefore, the expression profile measured at each spot can be treated as an independent observation. This approach gives a sufficient sample size for statistical power to detect differences in gene expression (Vickovic et al., 2019). Statistical analyses of spots as sample sizes must account for spatial autocorrelation of gene expression levels and genes with statistically significant spatial patterns (Stahl et al., 2016). Spatial autocorrelation is a statistical phenomenon that occurs when observations within a spatially defined region are more similar than observations from other regions. For our spatial transcriptomics, gene expression levels measured at adjacent spots on the Visium slide are likely to be more similar to each other than to spots that are further apart. By accounting for spatial autocorrelation, we can depict each spot as an independent observation for accurate and precise transcriptomic analysis.

Include species and sex of animals in the abstract.

This is included now in the abstract of the revised manuscript.

It would be helpful to show heatmaps of a few key DEGs superimposed on the histology images using Seurat, especially the genes specific to stratum radiatum and/or oriens (eg. Sgk1). Visual representations of the gene expression changes would emphasize the novelty and utility of this technique.

We included heat maps to illustrate the upregulation of *Sgk1* in stratum radiatum and oriens of CA1 (**Figure 5b**). This finding was further validated by *in situ* hybridization from an independent cohort of mice (**Figure 5c-f**). To further expand the novelty of this finding, we added new single nuclei multiomics data showing that the expression of *Sgk1* is induced within oligodendrocytes of stratum radiatum and oriens (**Figure 5g-l**).

• *For comparing bulk-RNA-seq of whole dorsal hippocampal tissue versus “pseudobulk” analysis from Visium, perhaps there would be a greater overlap of genes if the entire region of the hippocampus had been analyzed (including SR & SO of CA2/3 and hilus of the DG)? This would better mimic the amount of tissue analyzed with a bulk dissection of hippocampus.*

Pseudobulk analysis of whole hippocampus, including stratum radiatum and oriens of CA2/3 and hilus of the DG, did not improve the overlap of genes with bulk-RNA-seq. This could be due to the use of a 10 uM coronal brain slice compared to the entire dorsal hippocampus tissue.

Response to Reviewer Figure 1. Pseudobulk analysis from whole hippocampal region. **a**, Representative brain image of whole hippocampus selected for gene expression analysis. **b**, Venn diagram of upregulated genes in whole hippocampus pseudobulk vs bulk RNA-seq. **c**, Venn diagram of downregulated genes in whole hippocampus pseudobulk vs bulk RNA-seq.

Please define “pseudobulk” analysis in greater detail.

“Pseudobulk” analysis refers to the grouping of each spot on a Visium slice into a single pseudo-sample to resemble bulk tissue for RNA sequencing.

Line 394: For others using 10X Visium platform, it would be helpful to report the mean number of reads per spot and genes per spot, as sequencing depth varies depending on the number of spots covered by tissue within each frame.

A supplemental table (Extended data table 3) with these quality control metrics is in the revised manuscript.

You report that there were 8 brains frozen for spatial transcriptomics – does this mean you used 2 slides with 4 frames each? Please clarify.

This is correct; we used 2 slides with 4 frames each, 1 slice per animal (4 HC vs 4 SOR). We updated this information in the methods section of the revised manuscript.

Please consider commenting on the differences in sequencing technologies and how they are analyzed when discussing the results of bulk RNA-seq versus pseudobulk spatial transcriptomics.

The following paragraph was added to the discussion section to address this comment:

Although we observed an overlap in DEGs, how the libraries were prepared, the amount of tissue and subregions sequenced, and statistical approach for each technique explains why we observe differences in gene expression from these two techniques. We identified 105 more DEGs by bulk RNA-seq possibly because this approach included all RNA, except ribosomal RNA, within all cell types and subregions of the entire dorsal hippocampus. Pseudobulk spatial transcriptomics included the poly-A mRNAs in barcoded dots selected within principal cell layers, stratum radiatum and oriens on a 10 µm brain slice. Any additional differences in overlap can be attributed to the type of statistical test used to compute significant changes in gene expression. Pseudobulk was analyzed by rank-sum Kruskal-Wallis test, while bulk RNA-seq was analyzed by the conventional EdgeR statistical test. We could not use the same statistical test because EdgeR utilizes the mean instead of the median to compute fold-change and may increase the number of outliers or false positives for pseudobulk analysis. Therefore, some DEGs in bulk RNA-seq appeared as a lower fold-change in pseudobulk because they did not surpass the 1.4 threshold for the Kruskal-Wallis test, yet they still displayed differential expression following learning.

Check verb tenses for consistency across manuscript, for example, line 133 “comprised” instead of present test “comprise”.

- *Line 190 - this sentence does not refer to Figure 4c-d.*
- *Lines 211-212 : please define “Abeta” as amyloid beta*
- *Line 323: provide examples of the non-neuronal genes affected by learning*
- *Line 437: ‘polled libraries’? or ‘pooled’?*
- *Line 454: typo, “GO Tern Fusion” should be Term*

Thank you for these suggestions. We have revised the manuscript accordingly.

Consider showing in situ hybridization for the upregulated Nr4A, without and after learning. It would really make the authors’ case that it is a key upregulated gene, and would give information on what neurons are expressing it. Also, are the neurons expressing any of the Nr4as generally the same neurons expressing Arc or Fos, for example?

We provide new RNAscope *in situ* hybridization data showing learning-induced upregulation of *Nr4a1* and *Nr4a2* in CA1 but not in DG (**Supplemental Figure 5a-f**). We also found both *Nr4a1* and *Nr4a2* are induced in *Arc*-expressing CA1 neurons (**Supplemental Figure 5g-j**). This supports our findings from spatial transcriptomics that *Nr4a1* and *Nr4a2* are expressed within CA1, but not DG, and have a functional role in CA1 for memory consolidation.

Clarify HCC: were the animals handled like in the SOR? Were they taken out and put back in the HC?

All the animals (7 HC vs 7 SOR) were handled daily for 5 consecutive days in the SOR behavior room prior to training sessions. During handling, each animal was taken out of its cage and left on the palm of the experimenter’s hand for 2 minutes.

References

- Bahl E, C. S., Elsadany M, Vanrobaeys Y, Lin L-C, Giese KP, Abel T, Michaelson JJ. (2022). NEUROeSTIMator: Using Deep Learning to Quantify Neuronal Activation from Single-Cell and

- Spatial Transcriptomic Data. *bioRxiv*.
<https://doi.org/https://doi.org/10.1101/2022.04.08.487573>
- Chatterjee, S., Bahl, E., Mukherjee, U., Walsh, E. N., Shetty, M. S., Yan, A. L., Vanrobaeys, Y., Lederman, J. D., Giese, K. P., Michaelson, J., & Abel, T. (2022). Endoplasmic reticulum chaperone genes encode effectors of long-term memory. *Sci Adv*, 8(12), eabm6063.
<https://doi.org/10.1126/sciadv.abm6063>
- Hawk, J. D., Bookout, A. L., Poplawski, S. G., Bridi, M., Rao, A. J., Sulewski, M. E., Kroener, B. T., Manglesdorf, D. J., & Abel, T. (2012). NR4A nuclear receptors support memory enhancement by histone deacetylase inhibitors. *J Clin Invest*, 122(10), 3593-3602.
<https://doi.org/10.1172/JCI64145>
- Hock, B. J., Jr., & Bunsey, M. D. (1998). Differential effects of dorsal and ventral hippocampal lesions. *J Neurosci*, 18(17), 7027-7032. <https://doi.org/10.1523/JNEUROSCI.18-17-07027.1998>
- Kwapis, J. L., Alaghband, Y., Kramar, E. A., Lopez, A. J., Vogel Ciernia, A., White, A. O., Shu, G., Rhee, D., Michael, C. M., Montellier, E., Liu, Y., Magnan, C. N., Chen, S., Sassone-Corsi, P., Baldi, P., Matheos, D. P., & Wood, M. A. (2018). Epigenetic regulation of the circadian gene *Per1* contributes to age-related changes in hippocampal memory. *Nat Commun*, 9(1), 3323.
<https://doi.org/10.1038/s41467-018-05868-0>
- Liu, Y., Zhou, J., & White, K. P. (2014). RNA-seq differential expression studies: more sequence or more replication? *Bioinformatics*, 30(3), 301-304. <https://doi.org/10.1093/bioinformatics/btt688>
- Moser, E., Moser, M. B., & Andersen, P. (1993). Spatial learning impairment parallels the magnitude of dorsal hippocampal lesions, but is hardly present following ventral lesions. *J Neurosci*, 13(9), 3916-3925. <https://doi.org/10.1523/JNEUROSCI.13-09-03916.1993>
- Moser, M. B., Moser, E. I., Forrest, E., Andersen, P., & Morris, R. G. (1995). Spatial learning with a minislab in the dorsal hippocampus. *Proc Natl Acad Sci U S A*, 92(21), 9697-9701.
<https://doi.org/10.1073/pnas.92.21.9697>
- Pevzner, A., Miyashita, T., Schiffman, A. J., & Guzowski, J. F. (2012). Temporal dynamics of Arc gene induction in hippocampus: relationship to context memory formation. *Neurobiol Learn Mem*, 97(3), 313-320. <https://doi.org/10.1016/j.nlm.2012.02.004>
- Schurch, N. J., Schofield, P., Gierlinski, M., Cole, C., Sherstnev, A., Singh, V., Wrobel, N., Gharbi, K., Simpson, G. G., Owen-Hughes, T., Blaxter, M., & Barton, G. J. (2016). How many biological replicates are needed in an RNA-seq experiment and which differential expression tool should you use? *RNA*, 22(6), 839-851. <https://doi.org/10.1261/rna.053959.115>
- Stahl, P. L., Salmen, F., Vickovic, S., Lundmark, A., Navarro, J. F., Magnusson, J., Giacomello, S., Asp, M., Westholm, J. O., Huss, M., Mollbrink, A., Linnarsson, S., Codeluppi, S., Borg, A., Ponten, F., Costea, P. I., Sahlen, P., Mulder, J., Bergmann, O., . . . Frisen, J. (2016). Visualization and analysis of gene expression in tissue sections by spatial transcriptomics. *Science*, 353(6294), 78-82.
<https://doi.org/10.1126/science.aaf2403>
- Vazdarjanova, A., McNaughton, B. L., Barnes, C. A., Worley, P. F., & Guzowski, J. F. (2002). Experience-dependent coincident expression of the effector immediate-early genes *arc* and *Homer 1a* in hippocampal and neocortical neuronal networks. *J Neurosci*, 22(23), 10067-10071.
<https://doi.org/10.1523/JNEUROSCI.22-23-10067.2002>
- Vickovic, S., Eraslan, G., Salmen, F., Klughammer, J., Stenbeck, L., Schapiro, D., Aijo, T., Bonneau, R., Bergenstrahle, L., Navarro, J. F., Gould, J., Griffin, G. K., Borg, A., Ronaghi, M., Frisen, J., Lundeberg, J., Regev, A., & Stahl, P. L. (2019). High-definition spatial transcriptomics for in situ tissue profiling. *Nat Methods*, 16(10), 987-990. <https://doi.org/10.1038/s41592-019-0548-y>
- Vogel-Ciernia, A., & Wood, M. A. (2014). Examining object location and object recognition memory in mice. *Curr Protoc Neurosci*, 69, 8 31 31-17. <https://doi.org/10.1002/0471142301.ns0831s69>

REVIEWERS' COMMENTS

Reviewer #1 (Remarks to the Author):

The authors have done a thorough and thoughtful job addressing my previous comments. The inclusion of the Nr4a manipulation in DG significantly strengthens the manuscript.

I do have two minor remaining concerns due to the new data added since the first submission.

1) Please note in the methods the ITI for the three training trials used for SOR.

2) While the single cell RNA seq and single cell ATAC seq datasets are powerful, it seems a bit of a waste to use them just to confirm Sgk1 is increased in oligodendrocytes specifically, something that could be demonstrated with immuno staining of specific cell types. Additionally, for the sn ATAC-seq, the differences between the SOR and HC groups look quite minor. Also, the legend for 5i seems incorrect and is also the caption for 5i (“Dot plot showing expression of Sgk1 across all the cell types between homecage and learning”).

Overall, however, this is an exciting demonstration of the power of spatial transcriptomics to understand the mechanisms underlying memory consolidation and clarifying these minor points will make the manuscript suitable for publication.

Reviewer #2 (Remarks to the Author):

We are thoroughly impressed with the improvements made in response to the original feedback provided to the authors. The revised manuscript has undoubtedly been strengthened, and the additions made have substantially enhanced the scientific value of the study. In response to the concerns raised regarding the novelty of the study, we appreciate the authors' thorough clarification on the areas that contribute novel information to the literature. The points they highlighted indeed address some of the previous concerns.

We commend the dentate gyrus-specific manipulation of Nr4a transcription function using the AAV-based approach to assess long-term memory (Supplementary Figure 8). This new experimental data provides crucial functional relevance to the selective role of Nr4a family members in CA1, which is supported by the spatial transcriptomics experiment.

The integration of single-nuclei transcriptomics (using a Split-pool barcoding approach) and epigenomics (snATAC-seq) data has further enriched the scope of the research. The identification of Sgk1 upregulation in oligodendrocytes after learning, as revealed by the single-nuclei transcriptomics experiment (Figure 5g-i), has shed light on the role of these cells in memory consolidation. Moreover, the single-nuclei epigenomics data, demonstrating increased promoter accessibility of Sgk1 in

oligodendrocytes after learning (Figure 5j-l), provides further evidence of the functional significance of these cells in the memory consolidation process. The subsequent RNAscope validation of stratum radiatum- and oriens-specific upregulation of Sgk1 (Figure 5c-f) reinforces the reliability of your spatial transcriptomic approach.

The addition of low-throughput gene expression data at various time points after learning (Figure 2d) has enhanced the temporal resolution of the study. This data, combined with the quadrant plot comparing the transcriptomic profiles between the two batches of spatial gene expression experiments (Supplementary Figure 2), ensures the robustness and reproducibility of the findings.

Furthermore, the inclusion of the heatmap showing the expression levels of genes that are upregulated in two or more subregions after learning (Figure 3g) provides a clearer picture of the coordinated transcriptional changes occurring across hippocampal subregions during memory consolidation. This in-depth analysis highlights the interconnectedness of these subregions in the memory consolidation process, adding important insights to the study.

Regarding the assertion made in the authors response that manipulation of the ventral hippocampus has no effects on long-term spatial memory, I must bring to their attention some relevant studies that indicate otherwise: (Spellman et al., 2015, Nature), (Aqrabawi and Kim, 2018, Nature Communications), (Avigan et al., 2020, Hippocampus), etc. While not an exhaustive list, there is abundant evidence of ventral hippocampal subregions involvement in spatial learning and memory. Considering these studies, we believe that exploring the differences in gene expression patterns between the dorsal and ventral hippocampus could indeed provide valuable and novel insights into the transcriptional events that occur during memory consolidation. However, we also understand that such an investigation may extend beyond the scope of the current study, which focuses specifically on the dorsal hippocampus. The authors efforts to clarify their focus on the dorsal hippocampus in the revised manuscript is satisfactory.

We would also like to acknowledge the authors' general efforts in improving the clarity, readability, and overall presentation of the manuscript. The additions of new details, enhanced descriptions, and thorough explanations have strengthened the confidence in their claims. The manuscript now offers a more accessible and compelling narrative.

Overall, we want to reiterate our general appreciation for the improvements made to the manuscript. The addition of new data and analyses, as well as the detailed descriptions of the methodology and results, has significantly strengthened the study. We believe the manuscript is now well-prepared for publication in Nature Communications and will make a valuable contribution to the field.

Reviewer #3 (Remarks to the Author):

The authors have addressed all of my previous concerns and I have no new ones. In my opinion, they

have also adequately addressed the concerns of the other reviewers, some of which were valid issues. The revised manuscript provides important new information and will be a terrific resource for those in the field as well as others interested in the technique.

We would like to thank the editor and all three reviewers for having taken the time to thoroughly review our revised manuscript. We are gratified that our revisions were deemed satisfactory in addressing the reviewers' concerns.

A point-by-point response to all reviewers' comments

Note: The reviewer's comments are in *italics* and our answers are in blue.

Reviewer#1 (Remarks to the Author):

The authors have done a thorough and thoughtful job addressing my previous comments. The inclusion of the Nr4a manipulation in DG significantly strengthens the manuscript.

We thank the reviewer for appreciating our findings in the manuscript.

I do have two minor remaining concerns due to the new data added since the first submission.

1) Please note in the methods the ITI for the three training trials used for SOR.

We thank the reviewer for pointing this out. The Inter-trial Interval (ITI) in-between the training sessions of the SOR experiment was 5 minutes. We have included this statement in the Methods section of the manuscript.

2) While the single cell RNA seq and single cell ATAC seq datasets are powerful, it seems a bit of a waste to use them just to confirm Sgk1 is increased in oligodendrocytes specifically, something that could be demonstrated with immuno staining of specific cell types.

The single nuclei RNA-seq and single nuclei ATAC-seq experiments were originally performed in response to reviewer 2's recommendation to incorporate other techniques, such as epigenomics, in this study. The discrete spatial expression profile of Sgk1 was one of the most exciting findings from our spatial transcriptomics experiment. By incorporating the single-nuclei transcriptomics and epigenomics datasets, we were able to demonstrate a unique, cell-type-specific signature of Sgk1 expression after learning. Acquiring unbiased transcriptomic and epigenomic information at a single-cell resolution extends beyond the scope of conventional immunostaining methodologies. As rightly stated by reviewer #1, we do understand that these datasets contain additional information beyond Sgk1 expression, which will be examined in future studies.

Additionally, for the sn ATAC-seq, the differences between the SOR and HC groups look quite minor.

Among all the cell types examined using single nuclei ATAC-seq, only oligodendrocytes showed statistically significant chromatin accessibility on the promoter after learning (adjusted p-value 2.60E-22). We revised Figure 5I by including a heat map that depicts the extent of Sgk1 promoter accessibility across all cell types for better visualization of snATAC-seq data.

Also, the legend for 5I seems incorrect and is also the caption for 5i ("Dot plot showing expression of Sgk1 across all the cell types between homecage and learning").

We thank the reviewer for pointing out this error. The legends for Figure 5 are corrected in the revised manuscript.

Overall, however, this is an exciting demonstration of the power of spatial transcriptomics to understand the mechanisms underlying memory consolidation and clarifying these minor points will make the manuscript suitable for publication.

We appreciate the reviewer for the positive remarks on our study.

Reviewer #2 (Remarks to the Author):

We are thoroughly impressed with the improvements made in response to the original feedback provided to the

authors. The revised manuscript has undoubtedly been strengthened, and the additions made have substantially enhanced the scientific value of the study. In response to the concerns raised regarding the novelty of the study, we appreciate the authors' thorough clarification on the areas that contribute novel information to the literature. The points they highlighted indeed address some of the previous concerns.

We commend the dentate gyrus-specific manipulation of Nr4a transcription function using the AAV-based approach to assess long-term memory (Supplementary Figure 8). This new experimental data provides crucial functional relevance to the selective role of Nr4a family members in CA1, which is supported by the spatial transcriptomics experiment.

The integration of single-nuclei transcriptomics (using a Split-pool barcoding approach) and epigenomics (snATAC-seq) data has further enriched the scope of the research. The identification of Sgk1 upregulation in oligodendrocytes after learning, as revealed by the single-nuclei transcriptomics experiment (Figure 5g-i), has shed light on the role of these cells in memory consolidation. Moreover, the single-nuclei epigenomics data, demonstrating increased promoter accessibility of Sgk1 in oligodendrocytes after learning (Figure 5j-l), provides further evidence of the functional significance of these cells in the memory consolidation process. The subsequent RNAscope validation of stratum radiatum- and oriens-specific upregulation of Sgk1 (Figure 5c-f) reinforces the reliability of your spatial transcriptomic approach.

The addition of low-throughput gene expression data at various time points after learning (Figure 2d) has enhanced the temporal resolution of the study. This data, combined with the quadrant plot comparing the transcriptomic profiles between the two batches of spatial gene expression experiments (Supplementary Figure 2), ensures the robustness and reproducibility of the findings.

Furthermore, the inclusion of the heatmap showing the expression levels of genes that are upregulated in two or more subregions after learning (Figure 3g) provides a clearer picture of the coordinated transcriptional changes occurring across hippocampal subregions during memory consolidation. This in-depth analysis highlights the interconnectedness of these subregions in the memory consolidation process, adding important insights to the study.

We sincerely thank the reviewer for the appreciation and positive comments on our work.

Regarding the assertion made in the authors response that manipulation of the ventral hippocampus has no effects on long-term spatial memory, I must bring to their attention some relevant studies that indicate otherwise: (Spellman et al., 2015, Nature), (Agrabawi and Kim, 2018, Nature Communications), (Avigan et al., 2020, Hippocampus), etc. While not an exhaustive list, there is abundant evidence of ventral hippocampal subregions involvement in spatial learning and memory. Considering these studies, we believe that exploring the differences in gene expression patterns between the dorsal and ventral hippocampus could indeed provide valuable and novel insights into the transcriptional events that occur during memory consolidation. However, we also understand that such an investigation may extend beyond the scope of the current study, which focuses specifically on the dorsal hippocampus. The authors efforts to clarify their focus on the dorsal hippocampus in the revised manuscript is satisfactory.

We thank the reviewer for acknowledging our rationale to focus on the dorsal hippocampus in this study. We agree that exploring the transcriptomic diversity in the ventral hippocampus during spatial memory consolidation would be an exciting prospect for future studies. We have discussed this future direction and cited the references provided by the reviewer in the discussion section of the manuscript.

We would also like to acknowledge the authors' general efforts in improving the clarity, readability, and overall presentation of the manuscript. The additions of new details, enhanced descriptions, and thorough explanations have strengthened the confidence in their claims. The manuscript now offers a more accessible and compelling narrative.

Overall, we want to reiterate our general appreciation for the improvements made to the manuscript. The addition of new data and analyses, as well as the detailed descriptions of the methodology and results, has significantly strengthened the study. We believe the manuscript is now well-prepared for publication in Nature

Communications and will make a valuable contribution to the field.

We thank the reviewer for their thorough evaluation of the revised manuscript and their appreciative comments on the merits of our study.

Reviewer #3 (Remarks to the Author):

The authors have addressed all of my previous concerns and I have no new ones. In my opinion, they have also adequately addressed the concerns of the other reviewers, some of which were valid issues. The revised manuscript provides important new information and will be a terrific resource for those in the field as well as others interested in the technique.

We sincerely thank the reviewer for the appreciation and the supportive comments made on the revised manuscript.